# Loss of all three APP family members during development impairs synaptic function and plasticity, disrupts learning, and causes an autism-like phenotype

Vicky Steubler[1,†], Susanne Erdinger[1,†], Michaela K Back[2], Susann Ludewig[3,4], Dominique Fässler[1], Max Richter[1], Kang Han[1], Lutz Slomianka[5] (ID), Irmgard Amrein[5] (ID), Jakob von Engelhardt[2] (ID), David P Wolfer[5,6] (ID), Martin Korte[3,4] (ID) & Ulrike C Müller[1,*] (ID)

## Abstract

The key role of APP for Alzheimer pathogenesis is well established. However, perinatal lethality of germline knockout mice lacking the entire APP family has so far precluded the analysis of its physiological functions for the developing and adult brain. Here, we generated conditional APP/APLP1/APLP2 triple KO (cTKO) mice lacking the APP family in excitatory forebrain neurons from embryonic day 11.5 onwards. NexCre cTKO mice showed altered brain morphology with agenesis of the corpus callosum and disrupted hippocampal lamination. Further, NexCre cTKOs revealed reduced basal synaptic transmission and drastically reduced long-term potentiation that was associated with reduced dendritic length and reduced spine density of pyramidal cells. With regard to behavior, lack of the APP family leads not only to severe impairments in a panel of tests for learning and memory, but also to an autism-like phenotype including repetitive rearing and climbing, impaired social communication, and deficits in social interaction. Together, our study identifies essential functions of the APP family during development, for normal hippocampal function and circuits important for learning and social behavior.

**Keywords** Alzheimer; Amyloid precursor protein; Autism spectrum disorder; learning and memory; synaptic plasticity
**Subject Categories** Molecular Biology of Disease; Neuroscience
**The EMBO Journal (2021) 40: e107471**

## Introduction

Synaptic deficits and resulting impairments in neuronal circuits are hallmark features of neuropsychiatric disorders including autism and neurodegenerative diseases in particular Alzheimer's disease (AD; (Lin *et al*, 2016; Busche & Hyman, 2020)). In AD, synaptic dysfunction and loss of synapses occur early in the course of the disease and are highly correlated with disease severity and cognitive decline (Terry *et al*, 1991; Selkoe, 2002). As the precursor to the Aβ peptide that is deposited in the brains of AD patients, the amyloid precursor protein (APP) has been genetically and biochemically linked to AD more than 30 years ago (Goldgaber *et al*, 1987; Kang *et al*, 1987; Tanzi *et al*, 1987). Despite the assumed key role of Aβ for pathogenesis, numerous clinical trials directed at reducing Aβ have failed to substantially modify clinical symptoms or the course of the disease (Panza *et al*, 2019). Thus, it now appears crucial to understand AD pathogenesis in the context of APP physiological functions, particularly its role for brain morphology, synaptogenesis, synaptic plasticity, and behavior.

APP is a type I single-pass transmembrane protein that belongs to an evolutionary conserved gene family including APL-1 in *C. elegans*, APPL in *Drosophila*, Appa and Appb in *zebrafish*, and in mammals besides APP the amyloid precursor-like proteins APLP1 and APLP2 (reviewed in Müller *et al*, 2017; for recent examples on studies in orthologs, see Banote *et al*, 2020; Ewald & Li, 2012; Kessissoglou *et al*, 2020; Rieche *et al*, 2018; Wang *et al*, 2017). APP family proteins contain several regions of high sequence conservation including the short intracellular domain and the large extracellular N-terminus encompassing the E1 and E2 domains, with E1 mediating synaptic adhesion (Fig EV1A). Although APLPs lack the Aβ region, they undergo similar proteolytic processing by α-, β-, and

1   Department of Functional Genomics, Institute of Pharmacy and Molecular Biotechnology, Heidelberg University, Heidelberg, Germany
2   Institute of Pathophysiology, Focus Program Translational Neuroscience (FTN), University Medical Center of the Johannes Gutenberg University Mainz, Mainz, Germany
3   Division of Cellular Neurobiology, Zoological Institute, TU Braunschweig, Braunschweig, Germany
4   Helmholtz Centre for Infection Research, Neuroinflammation and Neurodegeneration Group, Braunschweig, Germany
5   Institute of Anatomy and Zurich Center for Integrative Human Physiology, University of Zurich, Zurich, Switzerland
6   Institute of Human Movement Sciences, ETH Zurich, Zurich, Switzerland
    *Corresponding author. Tel: +49 6221 54 6717; Fax: +49 6221 54 5830; E-mail: u.mueller@urz.uni-heidelberg.de
    †These authors contributed equally to this work

γ-secretases. There is evidence from both cultured neurons and engineered mouse models that APP and the APLPs are multimodal proteins mediating their functions via both the secreted ectodomains (e.g., APPsα) and the transmembrane full-length isoforms (Soba et al, 2005; Hick et al, 2015; Klevanski et al, 2015; Willem et al, 2015; Müller et al, 2017; Richter et al, 2018; Mockett et al, 2019; Rice et al, 2019). Despite recent progress, these physiological functions are still poorly understood.

Genetic evidence indicates partially overlapping functions within the gene family, with single APP-KO, APLP1-KO, or APLP2-KO mice being fully viable and exhibiting only subtle phenotypes (Heber, 2001; Li et al, 1996; Ring et al, 2007; Schilling et al, 2017; von Koch et al, 1997; Zheng et al, 1995), whereas germline double knockout (APP/APLP2-DKO and APLP2/APLP1-DKO) and APP/APLP1/APLP2 triple knockout (TKO) mice (von Koch et al, 1997; Heber et al, 2000; Herms et al, 2004) proved lethal shortly after birth due to deficits in neuromuscular transmission and morphology (Wang et al, 2005; Caldwell et al, 2013; Klevanski et al, 2014). The requirement of APP family proteins at the neuromuscular junction and the resulting lethality of constitutive TKO mice had so far precluded the analysis of the physiological functions of the APP family. Recently, a study reported on the generation of conditional TKO mice lacking the APP family in the adult forebrain (Lee et al, 2020). These adult cTKO mice revealed mild deficits in neuronal excitability and LTP that were associated with rather subtle impairments in the Morris water maze (Lee et al, 2020). Interestingly, and somewhat surprisingly, these impairments were considerably milder as compared to those of previously generated conditional APP/APLP2 double knockout mice (Hick et al, 2015). Given the importance and prominent expression of all APP family members during development and in particular during synaptogenesis (Wang et al, 2009; Weyer et al, 2014; Schilling et al, 2017), we set out to inactivate the APP gene family already during embryonic development to assess the full extent of APP family functions for brain development, synaptogenesis, and the mature central nervous system. To this end, we generated mice lacking APP and APLP2 in excitatory forebrain neurons from E11.5 onwards on a constitutive APLP1-KO background. These NexCre cTKO mice proved fully viable, but showed altered brain morphology with agenesis of the corpus callosum and impaired lamination of the hippocampus. Further, NexCre cTKO mice revealed impaired basal synaptic transmission and severely reduced long-term potentiation (LTP) that was associated with reduced spine density of hippocampal neurons. At the behavioral level, NexCre cTKO mice were not only severely impaired in several tests for learning and memory, but also exhibited core autism-like behaviors including stereotypic repetitive behaviors, reduced social communication, and impaired social interaction. Together, our study identifies essential functions of the APP family during development, for normal hippocampal function and circuits important for learning and social behavior.

# Results

## Conditional triple knockout mice lacking the entire APP family from embryonic stages onwards in excitatory forebrain neurons

To avoid the lethality of constitutive triple KO mice and to circumvent compensation by APLPs (for schematic overview of the APP family, see Fig EV1A), we generated mice with a conditional CNS specific APP/APLP1/APLP2 triple knockout (NexCre cTKO). To this end, we crossed APP$^{flox/flox}$/APLP2$^{flox/flox}$/APLP1$^{-/-}$ mice to NexCre mice (Goebbels et al, 2006) expressing Cre prenatally from about E11.5 onwards in postmitotic neuronal precursor cells of the cortex and hippocampus leading to APP and APLP2 gene deletion selectively in excitatory (but not inhibitory) forebrain neurons. Note that while APP and APLP2 alleles were floxed and deleted in a NexCre-specific pattern, APLP1 was constitutively inactivated in the germline (for breeding scheme, see Fig EV1B). Previously, we had shown that APLP1-KO mice, which we used as internal littermate (LM) controls in this study, exhibit a WT-like phenotype with regard to brain anatomy, basal synaptic transmission, and LTP, as well as spine density (Schilling et al, 2017). These NexCre cTKO mice were born at the expected Mendelian frequency (Fig EV1B) and proved viable up to at least 20 months of age (see Appendix).

## Impaired hippocampal lamination and agenesis of the corpus callosum in NexCre cTKO mice

Histopathological analysis of the brains of NexCre cTKO mice revealed no overt disruption of the laminar cytoarchitecture of the cerebral cortex but striking alterations in hippocampal structure (Fig 1). We used stereology and serial frontal sectioning of Giemsa-stained glycomethacrylate sections to estimate the volume of the neocortex in young adult (age: 5–6 months) and aged (age: 18–20 months) NexCre cTKO mice to assess possible age-dependent neurodegeneration. APLP1-KO littermates (LM, young and aged groups) and young wild-type (WT) mice served as controls. Total unilateral neocortical volume was estimated to be 32.9 mm$^3$ (SD 2.2, $n = 5$) in young wild-type mice, 32.8 mm$^3$ (SD 1.5, $n = 5$) in young LM controls, 34 mm$^3$ (SD 2.8, $n = 5$) in young cTKOs, 32.9 mm$^3$ (SD 3, $n = 5$) in aged LMs, and 33.2 mm$^3$ (SD 2, $n = 6$) in aged cTKOs. Estimates of the mean Gundersen–Jensen coefficient of error (CE) of repeated measures of cortical volume were < 0.01, i.e., measurements were very precise. Genotype effects or genotype × age interactions were not significant ($P > 0.42$ for all comparisons), indicating no difference in cortical volume with genotype or aging. The rostro-caudal neocortical distributions did not differ between genotypes and ages, with individual group bin means within the confidence intervals of the other group bin means (Fig 1A). In previously generated constitutive triple KO mice (Herms et al, 2004), we had observed focal neuronal ectopias in the marginal zone of the cortex, indicating over-migration of neuroblasts. Therefore, each analyzed section was inspected for gross abnormalities in cortical layering, signs of gliosis, and ectopias of which none were found in any section of any group. Likewise, no obvious alterations were found using layer-specific markers yielding comparable immunostaining for Ctip2 (layers V/VI) and calbindin (layers II/III and V) in the cortex of NexCre cTKOs and LM controls (Fig 1B).

In the dorsal (septal) hippocampus, CA1 pyramidal cells form a compact and continuous cell layer in WT and LM controls without any obvious sublayers (Fig 1C, see also magnification on the right). In young and aged NexCre cTKOs, the distal two-thirds of the septal pyramidal cell layer are split into deep and superficial sublayers with occasional interruptions of the sublayers along the transverse extent of the layer. Small ectopic groups of CA1 pyramidal cells

were found in the stratum radiatum (Fig 1C). In some NexCre cTKOs, this disruption extended into the proximal one-third of the layer. The expression of calbindin, as one marker expressed by superficial CA1 cells (Soltesz & Losonczy, 2018) was, despite the delamination, not affected in NexCre cTKOs. Superficial CA1 cells of cTKO mice still expressed calbindin, while deep CA1 cells did not (Fig 1C, right panel). Also the CA3 pyramidal cell layer of NexCre cTKOs appeared less compact than in LM or WT controls, with less

striking but clearly present alterations. Bilayering in NexCre cTKO extended rarely throughout the transverse extent of CA3 and often splits the layer diagonally (from superficial-proximal to deep-distal, see Fig 1C). In the ventral hippocampus, there were typically no or much less pronounced differences between NexCre cTKOs and LM or WT controls. We also estimated the total number of CA1 cells and took care to include ectopic groups of CA1 pyramids in NexCre cTKOs cell counts (Fig 1D). CA1 pyramidal cell numbers were

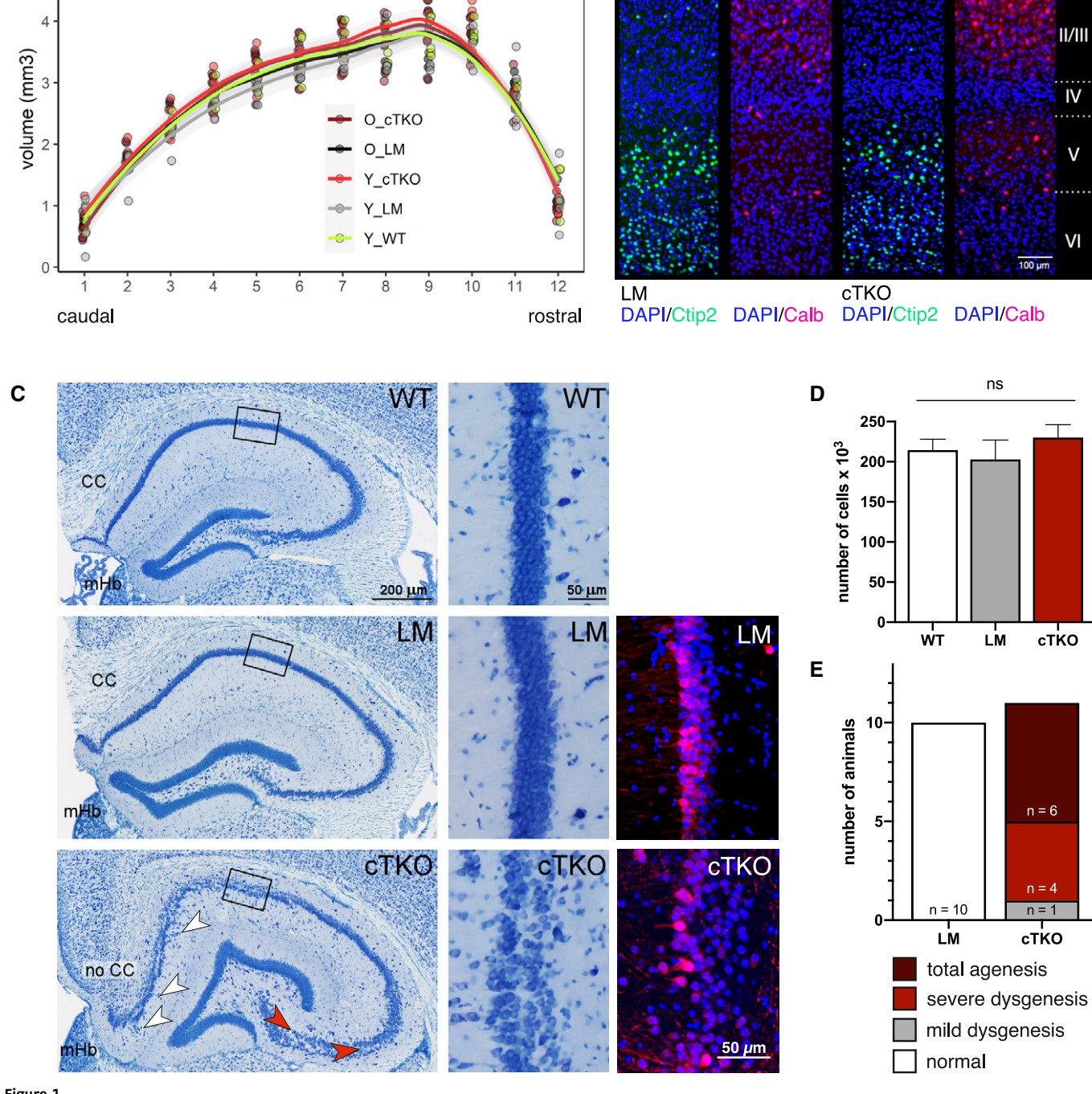

**Figure 1.**

**Figure 1.  NexCre cTKOs show agenesis of the corpus callosum and abnormal hippocampal lamination.**

A  Stereological evaluation of cortical volume of young (Y_WT, Y_LM, Y_cTKO, age: 5-6 months, each group *n* = 5) and old (O_LM *n* = 5, O_cTKO *n* = 6, age: 18–20 months) mice in 12 equidistant bins along the caudal-rostral axis of the coronally sectioned brain. Each bin would comprise an approximately 570-μm-thick slab of the brain. No significant differences between genotypes or age groups were seen in the volumes per bin or in the volume distributions along the caudal-rostral axis. Areas shaded in gray correspond to the 95% confidence intervals.

B  Double IHC of coronal brain sections (40 μm) of a littermate (LM) and a NexCre cTKO mouse (age: 5 months) stained for Ctip2 (green, expressed in layers V/VI) and calbindin (red, expressed in layers II/III and to some extent in layer V). Note that no gross abnormalities in cortical lamination were observed. Layers of DAPI-stained cell nuclei are denoted on the right (I-VI), scale bar: 100 μm.

C  Giemsa-stained, glycol methacrylate-embedded coronal brain sections (bregma −1.9) displaying the hippocampus and adjacent callosal fiber tracts of a wild type (WT, top), littermate control (LM, middle), and a NexCre cTKO (cTKO, bottom; age: 5–6 months). Note the agenesis of the corpus callosum (no CC) in NexCre cTKO mice, which should always be present at this bregma and in sections in which the medial habenula (mHb) is present (CC in WT and LM). While WTs and LMs show a compact layer of CA1 cells, NexCre cTKOs display a bilaminar cytoarchitecture (boxed regions marking higher magnification to the right) with scattered ectopic cells in the stratum radiatum (white arrowheads). In addition, the CA3 appears less compact (red arrowheads). Layer identity visualized via calbindin staining (right panel, scale bar: 50 μm).

D  Quantification of the number of CA1 cells in WT (white, *n* = 5, age: 5–6 months), LM (gray, *n* = 5, age: 18–20 months), and NexCre cTKO (red, *n* = 6, age: 18–20 months) mice. One-way ANOVA revealed no significant difference between genotypes (*P* = 0.0814). Data are represented as mean ± SD.

E  Quantification of callosal malformations. *n* = 10 LM and *n* = 11 NexCre cTKO animals.

estimated to be 214582 (SD 13526, *n* = 5) in young wild-type mice (5–6 months), 202862 (SD 24198, *n* = 5) in aged LM controls (18–20 months), and 230116 (SD 16128, *n* = 6) in aged cTKOs (18–20 months; Fig 1D). CEs of repeated estimates were estimated to 0.07 or lower. Cell numbers did not differ significantly between groups (one-way ANOVA, *P* = 0.0814), indicating the absence of neuronal loss with aging.

In addition, NexCre cTKO mice showed a high incidence of agenesis of the corpus callosum (CC). In WT and LM control mice, the CC was observed in 11–13 consecutive coronal sections. The corpus callosum was assessed and rated, according to the number of sections (see scheme in Fig EV1C) in which the CC was observed as normal (> 9 consecutive sections), mildly dysgenic (7–9 sections), severely dysgenic (< 7 sections), or agenic (0 sections). None of the littermate or WT controls showed a dysgenic CC (Fig 1C and E). Agenesis of the CC was found in the majority of NexCre cTKOs, the remainder being severely dysgenic with only one, mildly dysgenic exception (Fig 1E).

**Lack of APP family members in NexCre cTKO mice impairs neuronal morphology of CA1 pyramidal neurons**

Although hippocampal CA1 pyramidal neurons were thought for a long time to comprise a uniform population, recent evidence indicates differences in morphology, expression profile, connectivity, and function not only along the longitudinal and proximodistal axis of the hippocampus but also along the radial (deep to superficial) axis (reviewed in Geiller *et al*, 2017; Soltesz & Losonczy, 2018). Due to the cTKO-induced delamination of superficial (sCA1) and deep (dCA1) pyramidal cells in NexCre cTKO mice, we studied them separately (for scheme, see Fig 2A). For Sholl analysis, biocytin-filled neurons were imaged (Fig 2B left) and classified as either superficial (Fig 2A and B, sCA1 light blue) or deep (Fig 2A and B, dCA1 dark blue) and their dendritic tree was reconstructed (Fig 2C). We plotted the number of intersections (measured within circles centered on the soma, see Fig 2B right) against the distance from the soma. In this analysis, an increased number of intersections per Sholl sphere correspond to an increase in dendritic complexity. Due to their different connectivity, apical and basal dendrites were analyzed separately. When comparing dCA1 neurons of both genotypes (NexCre cTKO and LM), we detected no significant difference in total

dendritic length and no significant difference in the overall complexity of either basal or apical dendrites (Fig 2D, E, G and H). Only the number of primary basal dendrites (Fig 2F) was significantly reduced in dCA1 cells from NexCre cTKO mice (2.857 ± 0.173), as compared to littermate controls (3.857 ± 0.2515). More pronounced differences were found for sCA1 cells (Fig 2I–M). Total dendritic length of the apical dendrite of sCA1 cells (Fig 2M) was significantly reduced by about 15% in NexCre cTKO (2944 ± 160.2 μm) as compared to LM controls (3,465 ± 178.1 μm). Sholl analysis indicated a decrease in complexity in apical dendritic segments of sCA1 cells from NexCre cTKO mice in proximal dendritic regions at a distance of 150–180 μm and at 270 μm from the soma (Fig 2L). No significant difference was found for the total dendritic length (Fig 2J) of basal sCA1 dendrites. While Sholl analysis indicated no overall difference in dendritic complexity of basal dendrites (Fig 2I), the number of primary basal dendrites was significantly reduced in sCA1 cells of NexCre cTKOs (3.792 ± 0.366) as compared to LM mice (4.792 ± 0.3068; Fig 2K).

**NexCre cTKO dendrites show lamina-specific spine density deficits along the radial CA1 axis**

Next, we assessed spine density as a correlate of excitatory synapses, in either deep or superficial CA1 cells. Spine counts were performed separately in midapical portions of apical dendrites (a region densely innervated by CA3 cells) and in basal dendrites (Fig 3). Surprisingly, we found in NexCre cTKO mice a selective reduction in spine density in different dendritic regions of sCA1 (indicated in light blue) and dCA1 (indicated in dark blue) pyramidal neurons, respectively. While midapical segments of dCA1 cells were unaffected (Fig 3B), basal dendrites of dCA1 from NexCre cTKO cells showed a prominent reduction in spine density (Fig 3A, LM: 1.633 ± 0.051 spines/μm versus NexCre cTKO: 1.442 ± 0.044 spines/μm, **\*\*P* = 0.0067). In contrast, basal dendrites from superficial sCA1 cells of NexCre cTKO mice showed unaltered spine density (Fig 3D), whereas for this cell type, apical dendrites were severely impaired (Fig 3E, LM: 1.499 ± 0.037 spines/μm versus NexCre cTKO: 1.280 ± 0.044 spines/μm, **\*\*\*P* = 0.0005). Together, this indicates a region-specific selective loss of spines in either basal (dCA1) or apical dendrites (sCA1) of NexCre cTKO pyramidal cells along the radial axis of the CA1 band (Fig 3C and F).

## NexCre cTKO mice exhibit impaired synaptic function with deficits in basal synaptic transmission and long-term potentiation

Basal synaptic physiology was studied by recording AMPA receptor-mediated mEPSCs and GABA receptor-mediated mIPSCs in CA1

pyramidal cells (Fig 4A–J). To assess possible specific alterations in the two laminae of CA1 cells from NexCre cTKO mice (see Fig 2A), we analyzed dCA1 and sCA1 pyramidal cells separately. As compared to LM controls, dCA1 neurons from NexCre cTKO mice (depicted in dark blue, Fig 4A–E) showed slightly but significantly increased mEPSC and mIPSC peak amplitudes (mEPSCs: 9.41 ± 0.31

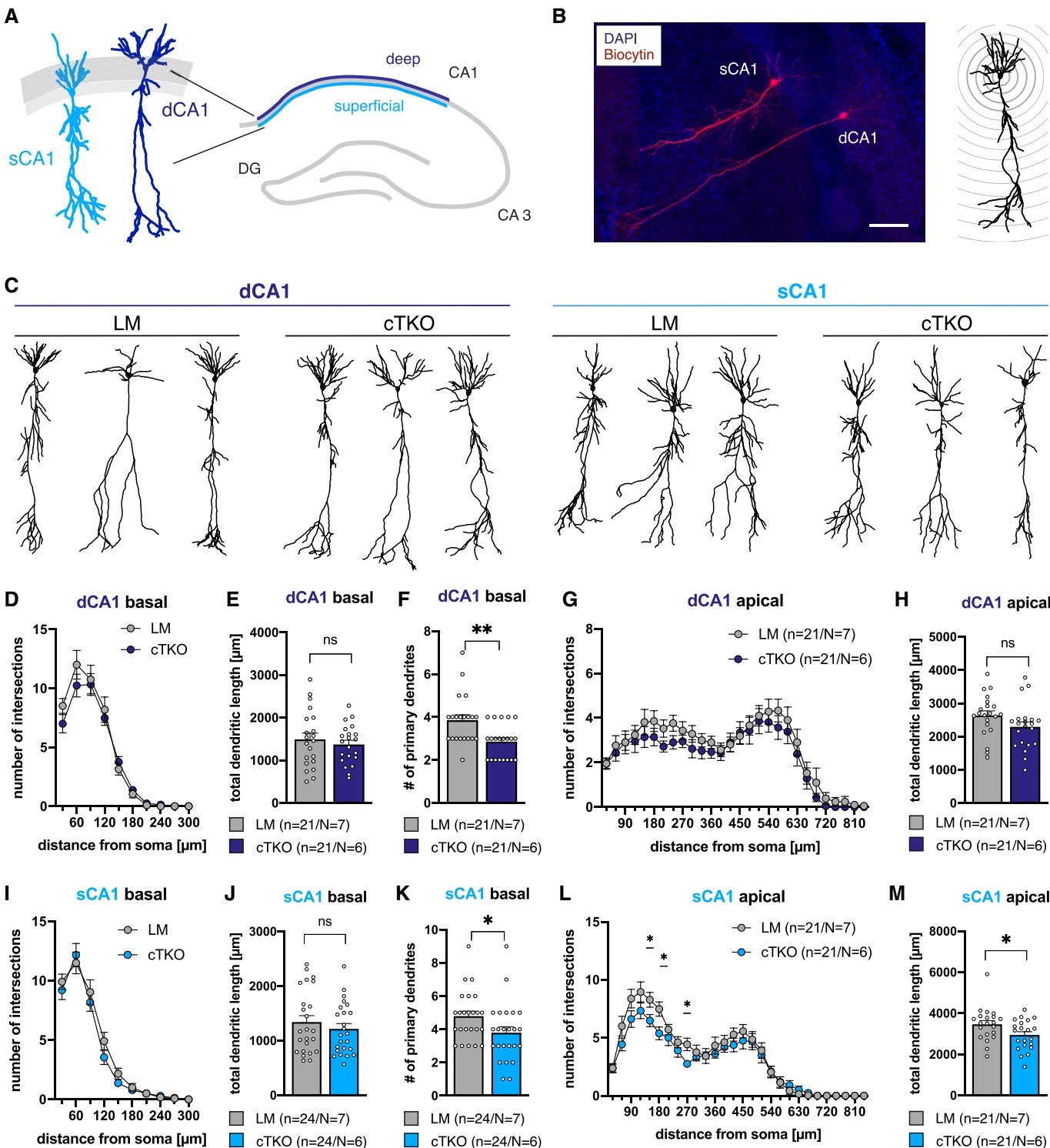

Figure 2.

**Figure 2.  Dendritic branching abnormalities in superficial and deep radial layers of CA1.**

A  Schematic representation of radial layers in the hippocampus.
B  Left: Biocytin-filled pyramidal cells in the CA1 region. DAPI staining of CA1 bundle allows differentiation between deep (right) and superficial (left) CA1 neurons (scale bar: 100 μm). Right: Schematic representation of Sholl spheres around the soma.
C  Representative 3D reconstructions of CA1 pyramidal neurons from deep (left, dark blue) and superficial (right, light blue) radial layers.
D  Sholl analysis reveals no genotype effect on basal dendritic segments of deep CA1 pyramidal neurons (dCA1; LM: $n = 21$, $N = 7$; cTKO: $n = 21/N = 6$).
E  Compared to littermate controls, dCA1 neurons of NexCre cTKO animals show no significant reduction in total dendritic length of basal dendrites (unpaired Student's $t$-test, $^{ns}$p = 0.4979).
F  The number of primary basal dendrites is significantly reduced in NexCre cTKO animals compared to LM in dCA1 cells (Mann–Whitney test, $**P = 0.0032$).
G  Sholl analysis reveals no genotype effect on apical dendritic segments of deep CA1 pyramidal neurons (dCA1; LM: $n = 21$, $N = 7$; cTKO: $n = 21/N = 6$).
H  Compared to littermate controls, dCA1 neurons of NexCre cTKO animals show no significant reduction in total dendritic length of apical dendrites (unpaired Student's $t$-test, $^{ns}$p = 0.1053).
I  Sholl analysis reveals no genotype effect on basal dendritic segments of superficial CA1 pyramidal neurons (sCA1; LM: $n = 24$, $N = 7$; cTKO: $n = 24/N = 6$).
J  Compared to littermate controls, sCA1 neurons of NexCre cTKO animals show no significant reduction in total dendritic length of basal dendrites (Mann–Whitney test, $^{ns}P = 0.5601$).
K  The number of primary basal dendrites is significantly reduced in NexCre cTKO animals compared to LM in sCA1 cells (Mann–Whitney test, $*P = 0.0232$).
L  Sholl analysis reveals a significant genotype effect on apical dendritic morphology of superficial CA1 pyramidal neurons (sCA1; two-way repeated measurements ANOVA: genotype $F(1, 40) = 4.643$, $*P = 0,0373$, with post hoc Sidak's multiple comparison test, $*P < 0.05$; LM: $n = 21$, $N = 7$; cTKO: $n = 21/N = 6$).
M  Compared to LM controls, sCA1 neurons of NexCre cTKO animals show a significantly reduced total dendritic length of apical dendrites (unpaired Student's $t$-test, $*P = 0.0358$).

Data information: $n$ = number of neurons, $N$ = number of animals (age: 4 months). Data are represented as mean ± SEM.

pA versus $11.2 \pm 0.63$ pA; mIPSCs: $8.36 \pm 0.25$ pA versus $9.37 \pm 0.36$ pA; Fig 4C and E), while mEPSC and mIPSC frequencies in dCA1 neurons were comparable between genotypes (Fig 4B and D). Whereas sCA1 neurons of NexCre cTKO mice (Fig 4F–J) had normal mEPSC amplitudes, we detected an about twofold increase in mEPSC frequency ($0.49 \pm 0.063$ Hz versus $0.89 \pm 0.13$ Hz, Fig 4G). mIPSC amplitude and frequency were unaltered in sCA1 neurons. To assess NMDA receptor functionality, we recorded AMPA- and NMDA receptor-mediated currents at a holding potential of $-70$ and $+40$ mV, respectively (Fig 4K and L). The amplitude of the NMDA receptor-mediated current was measured 35 ms after the stimulus artifact. The ratio of AMPA/NMDA receptor-mediated currents (A/N) in cTKO neurons did not significantly differ from that obtained in wild type or LM neurons, suggesting that the synaptic content of AMPA receptors and NMDA receptors is not grossly altered in cTKOs (Fig 4K and L). Overall, mEPSC and mIPSC recordings revealed rather mild changes in basal synaptic physiology.

To gain further insight and to assess basal synaptic transmission at the network level, we performed extracellular field recordings at Schaffer collateral axons projecting to CA1 apical dendrites. To assess first whether presynaptic components are altered and thus contributing to the observed increase in mEPSC frequency of sCA1 cells (see Fig 4G), we investigated short-term plasticity using the paired-pulse facilitation (PPF) paradigm (Fig 4M). Compared to LM controls, NexCre cTKO mice revealed significantly altered facilitation at high ($*P$(ISI 160ms) = 0.016) and very short ($*P$(ISI 10ms) = 0.047, unpaired Student's $t$-test) inter-stimulus intervals. This indicates that release probability is increased in Schaffer collateral/commissural fiber synapses and may explain why mEPSC frequency of sCA1 cells is increased although spine density is decreased. Recording of input–output (IO) curves (Fig 4N) or measuring IO strength at given fiber volley amplitudes (Fig 4O) both indicated a pronounced and highly significant reduction in fEPSP responses for all stimulus conditions. The amplitude of the fiber volley represents the number of axons firing an action potential and thus serves as an estimate of the strength of the afferent input that elicits smaller fEPSP responses in NexCre cTKOs.

To assess whether anatomical and morphological deficits of pyramidal cells are accompanied by alterations in synaptic and network activity of the hippocampus, we analyzed activity-dependent synaptic plasticity in NexCre cTKO mice (Fig 4P and Q). After 20 min of baseline activity recording, we induced long-term potentiation (LTP) at the CA3/CA1 (Schaffer collateral) pathway by application of theta burst stimulation (TBS) in acute hippocampal slices from adult NexCre cTKO mice or LM controls (age: 4–5 months). Of note, we had previously shown that mice of this age lacking only APLP1 do not exhibit any deficits in LTP or alterations in basal synaptic transmission (Schilling et al, 2017). Here, NexCre cTKO mice revealed a highly significant LTP deficit both after LTP induction (see Fig 4P; t20-25 LM control = $194.80 \pm 7.12\%$ versus NexCre cTKO = $128.04 \pm 4.71\%$, $P = 1.4*10^{-9}$) and for LTP maintenance. The average potentiation analyzed 55–60 min after TBS (t75–80, see Fig 4P and Q) in LM control slices amounted to $145 \pm 4.9\%$ ($n = 24$ slices from $N = 5$ mice), whereas potentiation was massively reduced in NexCre cTKOs to $109 \pm 3.4\%$ ($n = 23$ slices from $N = 5$ mice; unpaired Student's $t$-test $P = 5.72*10^{-7}$). We conclude that the overall impaired basal synaptic transmission at CA3/CA1 synapses leads to impaired synaptic plasticity in NexCre cTKO mice.

## NexCre cTKO mice exhibit hyperactivity in the home cage and impaired performance in learning and memory tasks

The drastic changes in synaptic plasticity observed in the hippocampus of NexCre cTKO mice suggested that hippocampus-dependent learning and memory formation is affected in these animals. As a baseline for subsequent cognitive tests, we first examined neuromotor performance of NexCre cTKO mice. Compared to LM controls, NexCre cTKO showed no alteration in grip strength and rotarod performance (Fig EV2A and B) indicating normal muscle strength and motor coordination. In line with this, NexCre cTKO showed no impairment in locomotion in the open field (OF), that assesses locomotor activity and exploratory behavior in a novel environment. Unlike controls, however, NexCre cTKO mutants did not habituate

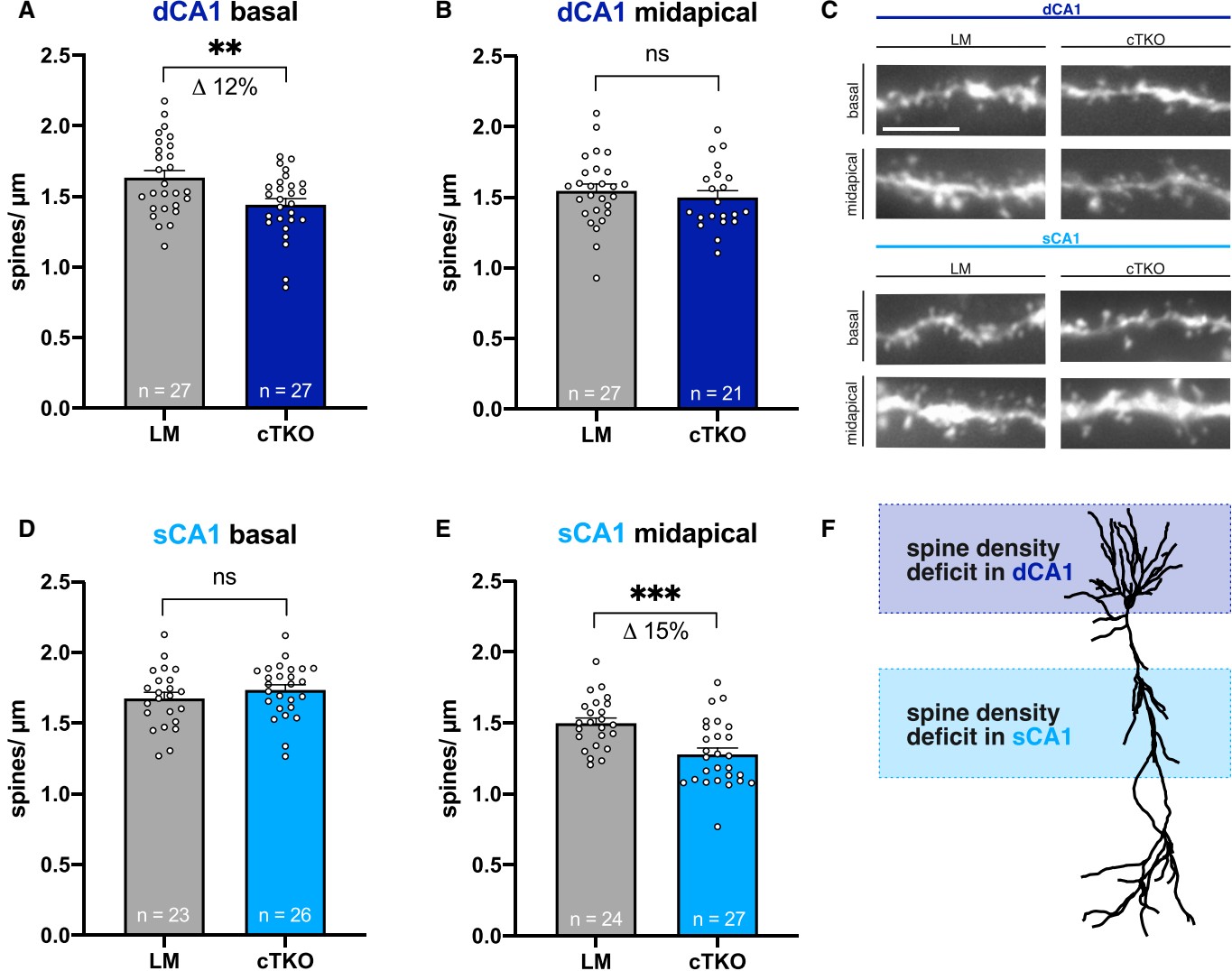

**Figure 3. Spine density is selectively affected along the radial axis of CA1.**

A, B   Spine density of deep CA1 pyramidal cells is significantly reduced in basal dendritic segments (A; unpaired Student's *t*-test, **P = 0.0067), while it is unchanged in midapical dendritic segments (B; unpaired Student's *t*-test, nsP = 0.4949).

C   Representative images of basal and midapical dendritic segments from both radial layers. Brightness and contrast were adjusted to ensure a uniform appearance. Scale bar: 5 μm.

D, E   Spine density of superficial CA1 pyramidal cells is normal in basal dendritic segments (D; unpaired Student's *t*-test, nsP = 0.3004), while it is significantly reduced in midapical dendritic segments (E; unpaired Student's *t*-test, ***P = 0.0005).

F   Schematic representation of affected regions dependent on position of the pyramidal cell along the radial axis of CA1.

Data information: *n* = number of neurons from 6-7 animals per genotype (age: 4 months). Data are represented as mean ± SEM.

but instead showed paradoxical hyperactivity during the second 5 min of testing on both consecutive days of testing (Fig EV2C). In addition, NexCre cTKO mutants showed strongly increased wall zone preference at the expense of both the transition and center zone (Fig EV2D). Overall, locomotion of NexCre cTKO mutants in the OF appeared stereotypical and mostly consisted of running along the wall (Fig EV2E). Diurnal activity profile in a familiar home cage was monitored in individual cages after habituation (Fig EV2F). Light phase activity was not significantly different between genotypes (Figs 5A and EV2F). While in LM control mice dark phase

activity declined toward the end of the dark phase, NexCre cTKO showed massively overshooting activity in the dark phase that was particularly pronounced at the end of the dark phase during day 1 and throughout the dark phase of day 2 (Fig 5A).

Next, we went on to study behavior in several hippocampus-dependent tasks including species-typic nesting and burrowing behavior, which is highly sensitive to hippocampal dysfunction (Deacon *et al*, 2002). In both tasks, NexCre cTKO mice showed major impairments (Fig 6A and B). To assess spatial working memory, mice underwent testing in the T-maze and the radial maze.

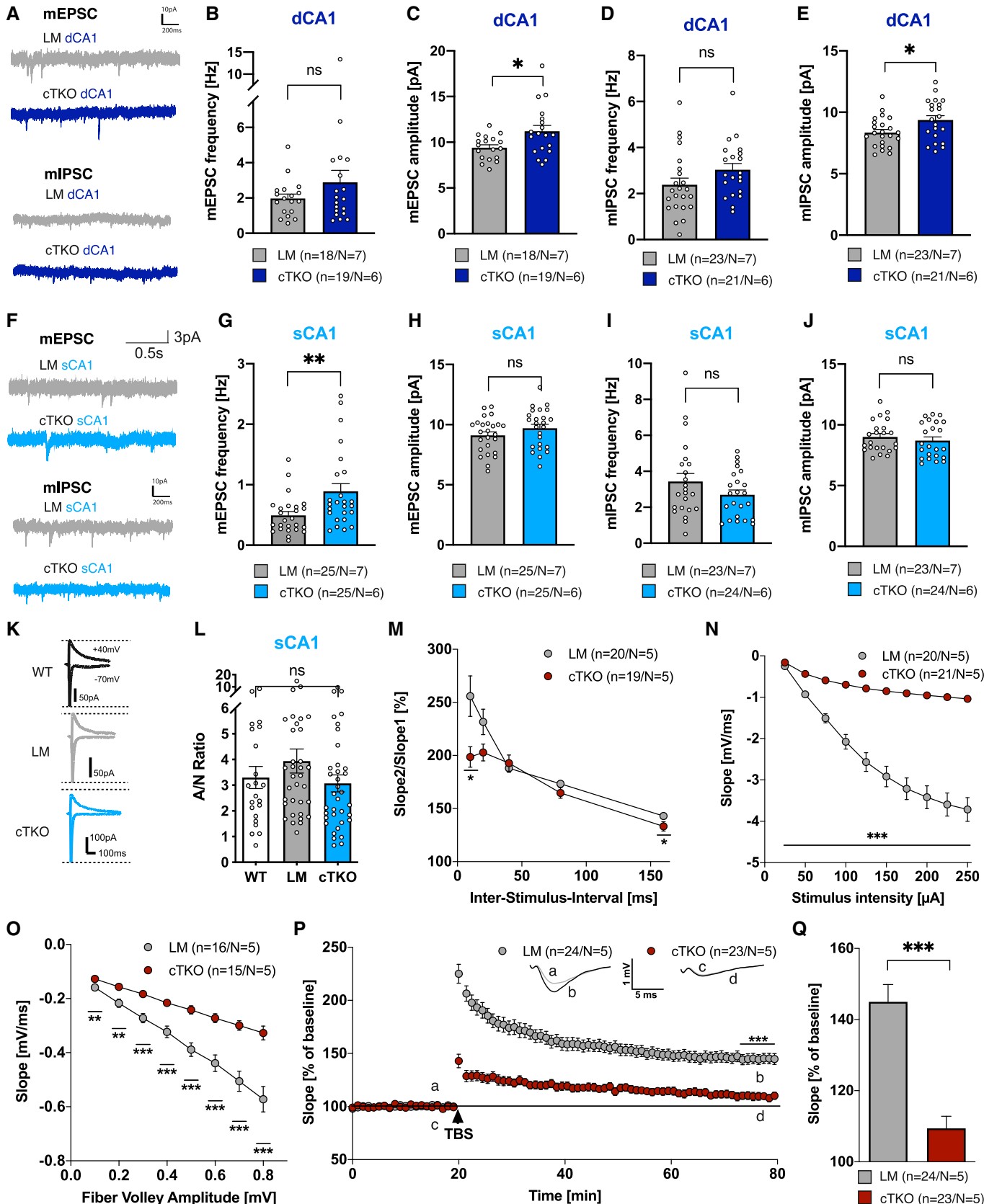

**Figure 4.**

**Figure 4.   NexCre cTKOs reveal severe impairments in basal synaptic transmission and synaptic plasticity.**

A   mEPSC and mIPSC sample traces of recordings from deep CA1 pyramidal cells.
B–E   Bar graphs of mEPSC and mIPSC frequencies and amplitudes in deep CA1 pyramidal cells. mEPSC frequency was not different in deep CA1 pyramidal neurons of adult (4 months) NexCre cTKO versus littermate controls (B, $P = 0.6415$). mEPSC amplitude was increased in deep CA1 pyramidal cells of NexCre cTKO compared to LM control cells (C, *$P = 0.0363$). mIPSC frequency was not different in deep CA1 pyramidal neurons of adult (4 months) NexCre cTKO versus littermate controls (D, $P = 0.0564$) and mIPSC amplitude was increased in deep CA1 pyramidal cells of NexCre cTKO compared to LM (E, *$P = 0.0356$).
F   mEPSC and mIPSC sample traces of recordings from superficial CA1 pyramidal cells.
G–J   mEPSC frequency was increased in superficial CA1 pyramidal neurons of adult (4 months) NexCre cTKO versus littermate controls (G, **$P = 0.0052$). mEPSC amplitude was not altered (H, $P = 0.1579$). mIPSC frequencies were not different in superficial CA1 pyramidal neurons of adult (4 months) NexCre cTKO versus littermate controls (I, $P = 0.3831$), and mIPSC amplitude was not altered in superficial CA1 pyramidal cells of NexCre cTKO mice compared to cells of LM (J, $P = 0.3712$).
K   Sample traces of AMPA and NMDA receptor-mediated currents recorded from CA1 pyramidal cells.
L   A/N ratio was not changed in CA1 pyramidal neurons of adult (4 months) cTKO (light blue, $n = 36$) versus littermate controls (LM, gray, $n = 34$) and wild-type animals (white, $n = 22$; Kruskal–Wallis test: $P = 0.2748$).
M   Paired-pulse stimulation revealed a significant defect of short-term plasticity at ISI 10 and 160 ms (unpaired Student's $t$-test, *$P < 0.05$).
N   The correlation of given stimulus intensities to fEPSP slope revealed a highly significant deficit of NexCre cTKO (red) compared to LM (gray) (unpaired Student's $t$-test, ***$P < 0.0001$).
O   The correlation of fEPSP slope to defined fiber volley amplitudes revealed a reduced IO strength in NexCre cTKO animals (unpaired Student's $t$-test, **$P < 0.01$, ***$P < 0.001$).
P, Q   LTP was induced after 20-min baseline activity recording at hippocampal CA3-CA1 synapses (arrowhead, TBS). TBS led to an overall increase of synaptic efficacy in LM controls ($145 \pm 4.9\%$, gray), which was significantly reduced in NexCre cTKO ($109 \pm 3.4\%$, red) 60 min after TBS (t75-80; unpaired Student's $t$-test, ***$P = 5.71562E-07$). The insets show original traces of representative individual experiments.

Data information: (A–L): $n$ = number of slices from 6 to 7 animals per genotype (age: 4 months). Data are represented as mean $\pm$ SEM and were analyzed by Mann–Whitney test, if not further indicated. (M–Q): $n$ = number of slices, $N$ = number of animals (age: 4–5 months). Data are represented as mean $\pm$ SEM.

In the T-maze, NexCre cTKO mice showed a very poor performance not significantly different from chance level (Fig 6C). Strongly impaired performance was also revealed upon testing NexCre cTKO mice in an 8-arm fully baited radial maze. Whereas LM controls learned the task very well, achieving on average 7 correct among the first 8 choices at the end of training, NexCre cTKOs made much less correct choices than controls throughout training (Fig 6D), with many NexCre cTKO mice performing below chance level. When collecting 8 baits from the 8 freely accessible arms of the radial maze, re-entries into arms whose bait had already been collected are counted as working memory errors. NexCre cTKO made significantly more re-entry errors throughout training (Fig 6E) and failed to improve over time. In both groups, the number of re-entry errors increased with the number of baits already collected, reflecting the increasing challenge of working memory, but in NexCre cTKO mice, this increase was much steeper than in LM controls (Fig 6F). Throughout training, NexCre cTKO mutants had a much stronger tendency than controls to repeatedly visit a preferred arm, explaining their persistent performance below chance level (Fig 6G, compare Fig 6D). Many NexCre cTKO mutants did not collect all baits, indicative of problems with motivation or learning of the basic task procedure (Fig 6H).

To study hippocampus-dependent spatial reference memory, mice underwent testing in the Barnes maze that consists of a brightly lit circular table with 20 circular holes around its circumference. Under one of the holes is an "escape box" which the mouse can reach through the corresponding hole on the table top. The test exploits the rodents' aversion to open spaces, motivating them to seek shelter in the escape box. Unlike LMs that rapidly learned to find the escape box across training trials, latency to escape into the goal box did not decrease in NexCre cTKO mutants (Fig 6I). On each day, there were mutant animals failing to escape on all four trials. In contrast to LM control mice, NexCre cTKO mutants also failed to reduce their number of errors with training (Fig 6J). Importantly, the fraction of training trials with a direct spatial strategy was strongly reduced in

NexCre cTKO mutants, in favor of mixed strategies including freezing in the start zone (Fig 6K). Time to leave the start zone in the maze center was strongly increased in cTKO mutants (Fig 6L). During the probe trial (24 h after the last training session), conducted to test spatial reference memory, NexCre cTKO mutants made few pokes without evidence of spatial selectivity for the goal position. In contrast, LM controls showed robust spatial retention (Fig 6M).

Also in the Morris water maze, NexCre cTKO mutants displayed an overall strongly impaired performance during both acquisition and reversal learning. While LM controls showed clear signs of learning, progressively reducing their escape latency over time, there was no evidence of learning in NexCre cTKO mice (Fig 6N). Swim path length of NexCre cTKO did not decrease over the whole training period (Fig 6O). The mildly reduced swim speed of NexCre cTKOs (Fig 6P) can only partially account for their performance deficits. During the probe trial (first trial after platform relocation), NexCre cTKO mice spent significantly less time in the target zone as compared to LM controls (Fig 6Q). Due to persisting excessive wall-hugging, NexCre cTKO mutants avoided the target as well as control zones in adjacent quadrants and, as a group, failed to show a significant preference for the trained target (Fig 6Q and R). To further study confounding behaviors that may prevent mutants to develop a spatial search strategy, mice were subsequently tested for cue navigation in the MWM. Although NexCre cTKO clearly improved across trials, excluding major impairments in vision, they were again severely impaired in comparison with LM controls (Fig EV3A). This is likely due to highly increased levels of thigmotaxis and pronounced floating (Fig EV3B and C), also observed during non-cued training. Some NexCre cTKO mutants showed unusual impulsivity and frequently jumped off the flagged platform, which was seen only very rarely in LM controls (Fig EV3D). In summary, NexCre cTKO showed a nearly complete disruption of learning in the radial maze, Barnes maze, and MWM. They also performed very poorly in species-typical behaviors but showed normal muscle function and coordination.

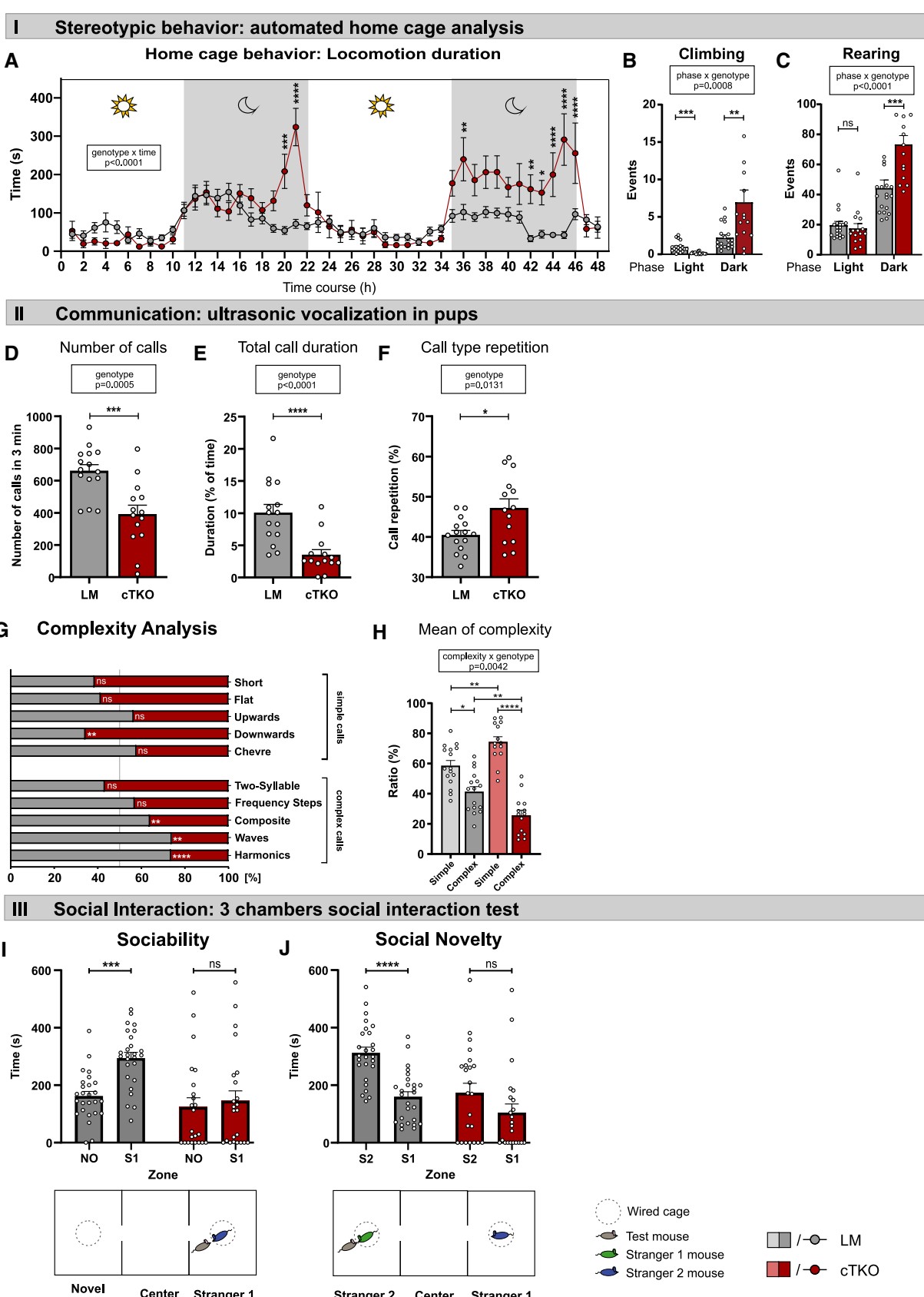

**Figure 5.**

**Figure 5.  NexCre cTKO animals show an autism spectrum disorder ASD-like phenotype with repetitive behavior and impairments in social communication and interaction.**

A   Diurnal behavior in the home cage (HC). NexCre cTKOs show overshooting activity in the dark phase. Line graphs illustrate the average locomotion per hour [Sidak's multiple comparison test, geno $F_{(1,32)}$ = 9.503, $P$ = 0.0042; timepoint $F_{(6.102,195.3)}$ = 10.51, $P$ < 0.0001; timepoint × geno $F_{(47,1504)}$ = 5.874, $P$ < 0.0001].

B   HC, climbing events are elevated in NexCre cTKO animals during the dark phase. During the light phase, NexCre cTKOs climb significantly less than LM controls [Sidak's multiple comparison test, geno $F_{(1,32)}$ = 6.218, $P$ = 0.0180; phase $F_{(1,32)}$ = 29.16, $P$ < 0.0001; phase × geno $F_{(1,32)}$ = 13.88, $P$ = 0.0008].

C   C, rearing events are elevated in NexCre cTKO animals during the dark phase [Sidak's multiple comparison test, geno $F_{(1,32)}$ = 6.565, $P$ = 0.0153; phase $F_{(1,32)}$ = 133.5, $P$ < 0.0001; phase × geno $F_{(1,32)}$ = 20.05, $P$ < 0.0001].

D   Ultrasonic vocalization (USV), NexCre cTKOs show a strong reduction of emitted calls compared to LM controls [unpaired Student's *t*-test ***$P$ = 0.0005].

E   USV, compared to LM controls, the duration of time spent (in %) with call emission is strongly reduced in NexCre cTKOs [Mann–Whitney test ****$P$ < 0.0001].

F   USV, NexCre cTKO animals repeat calls of a certain type more frequently [Welch's *t*-test *$P$ = 0.0131]

G   USV, analysis of call type complexity. NexCre cTKOs used simple calls with either similar or reduced frequency (downwards). Note that NexCre cTKOs used complex call types less frequently. Simple call types: short (Mann–Whitney test; ns, $P$ = 0.0747), flat (Mann–Whitney test; ns, $P$ = 0.2172), upwards (Mann–Whitney test; ns, $P$ = 0.2383), downwards (unpaired Student's *t*-test; LM $n$ = 1359 calls, NexCre cTKO $n$ = 1507 calls; **$P$ = 0.0085), and chevre (Welch's test; ns, $P$ = 0.1281). Complex call types are two syllable (Mann–Whitney test; ns, $P$ = 0.2380), frequency steps (Mann–Whitney test; ns, $P$ = 0.5246), composite (unpaired Student's *t*-test; LM = 2258 calls, NexCre cTKO $n$ = 722 calls; **$P$ = 0.0086), waves (Welch's test; LM = 403 calls, NexCre cTKO $n$ = 88 calls; **$P$ = 0.0039), and harmonics (Mann–Whitney test; LM = 711 calls, NexCre cTKO $n$ = 197 calls; ****$P$ < 0.0001).

H   USV, mean call type complexity. The ratio of complex calls is reduced in NexCre cTKOs in favor of simple call type emission [two-way ANOVA, complexity $F_{(1,27)}$ = 44.11, $P$ < 0.0001; complexity × geno $F_{(1,27)}$ = 9.751, $P$ = 0.0042].

I   Sociability in the three chambers test (3CT). LM controls spend more time with mouse stranger 1 (S1), whereas NexCre cTKOs spend comparable time exploring the novel object (NO) or S1. Bottom: Schematic representation of test [paired Student's *t*-test comparison between chambers LM ***$P$ = 0.0002; Wilcoxon test between chambers NexCre cTKO ns $P$ = 0.5412].

J   3CT, test for social novelty. Unlike LM controls, NexCre cTKO do not spend more time with an unfamiliar stranger mouse S2 compared to S1 [paired Student's *t*-test between chambers LM ****$P$ < 0.0001; Wilcoxon test between chambers NexCre cTKO ns $P$ = 0.1084].

Data information: (A-C): age: 3–4 months, $n$ = 19 LM and $n$ = 15 cTKO animals; (D-H): age: postnatal day 7, $n$ = 15 LM and $n$ = 14 cTKO animals; (I,J): age: 2 months, $n$ = 26 LM and $n$ = 23 cTKO animals. Balanced sex for all groups. Data are represented as mean ± SEM. *$P$ < 0.05, **$P$ < 0.01, ***$P$ < 0.001, ****$P$ < 0.0001, and ns not significant.

## NexCre cDKO mice show core autism-like behaviors including stereotypic repetitive behaviors, reduced social communication, and impaired social interaction

Repetitive stereotypic behaviors and behavioral inflexibility are key features of autism spectrum disorders (ASDs) in patients and ASD mouse models. Several observations prompted us to assess a possible autism-like phenotype in NexCre cTKO mice. During cognitive tests, we had observed not only very severe overall deficits in performance but also a high incidence of perseverative behaviors such as running along the wall in the OF, wall-hugging in the MWM, repetitive entries into one preferred arm during RAM testing, and frequent re-entry into the same arm of the T-maze.

In addition, NexCre cTKO mice showed very poor breeding performance, which was due to poor pregnancy rates and reduced survival of pups raised by NexCre cTKO mothers. From 54 breeding pairs consisting of LM males and NexCre cTKO females, only 50% (27 out of 54 females) got pregnant over a period of at least 2 months. When NexCre cTKO males were mated with LM females, pregnancy rates were further reduced and amounted to only 16% (16 out of 102 females). NexCre cTKO females frequently cannibalized newborn mice and showed reduced interest in their pups. From those newborn pups that survived the first 2 days after birth, NexCre cTKO females raised only 70% up to weaning (130 of 185 pups from 31 litters), whereas LM mothers raised 94% of their newborn pups (273 of 289 pups obtained from 41 litters). Genotype distribution in weaned offspring was as expected from Mendelian frequency, excluding embryonic or early postnatal lethality (EV1B). In a separate line, we had kept homozygous breeding pairs of APLP1-KO mice (same genotype as LMs), with continuous mating for 2 months each. From 50 APLP1-KO homozygous intercrosses, 94% got pregnant (47 of 50 females) and raised 94% of their pups

(829 of 878 from 121 litters). Together, these data suggest impaired mating behavior of NexCre cTKO mice and disturbed maternal/pup interaction for NexCre cTKO mothers. To circumvent these breeding problems, most animals of this study were generated by *in vitro* fertilization and embryo transfer into outbred foster mothers.

When recording general activity profiles in the home cage, we therefore also specifically assessed repetitive behaviors. Compared to LM controls, NexCre cTKO mice showed significantly increased climbing during the dark phase and a reduced number of climbing events during the light phase (Fig 5B), indicating hypersynchrony to the diurnal light cycle. Also rearing events were significantly increased in NexCre cTKOs (Fig 5C) during the dark, whereas no significant differences were observed for grooming. Combined with stereotypic behavior during cognitive testing, these data further corroborate an altered behavioral repertoire with a high tendency for repetitive behaviors.

Ultrasonic vocalization (USV) emission constitutes a major mode of communication in rodents (Wohr & Schwarting, 2013), and ASD mouse models are typically altered in USV production (Scattoni *et al*, 2009). When USVs were measured, NexCre cTKO pups (postnatal day P7) separated from their mothers emitted fewer USVs that were of shorter total duration (Fig 5D and E) and had a higher frequency as compared to LM controls (LM: 78.75 ± 1.16 kHz versus NexCre cTKO 83.52 ± 1.17 kHz, Mann–Whitney test: $P$ = 0.0059). As body weight of pups was comparable between genotypes (LM: 5.65 ± 0.11 g versus NexCre cTKO 5.33 ± 0.15 g, unpaired Student's *t*-test: ns), this excludes major differences in overall strength as a cause of impairments. Atypical patterns of vocalizations have been reported in infants with ASD (Esposito & Venuti, 2010) and ASD mouse models (Wohr *et al*, 2015; Takahashi *et al*, 2016; Eltokhi *et al*, 2018). Therefore, we analyzed the vocalization repertoire in more detail. Neonatal calls emitted upon pup

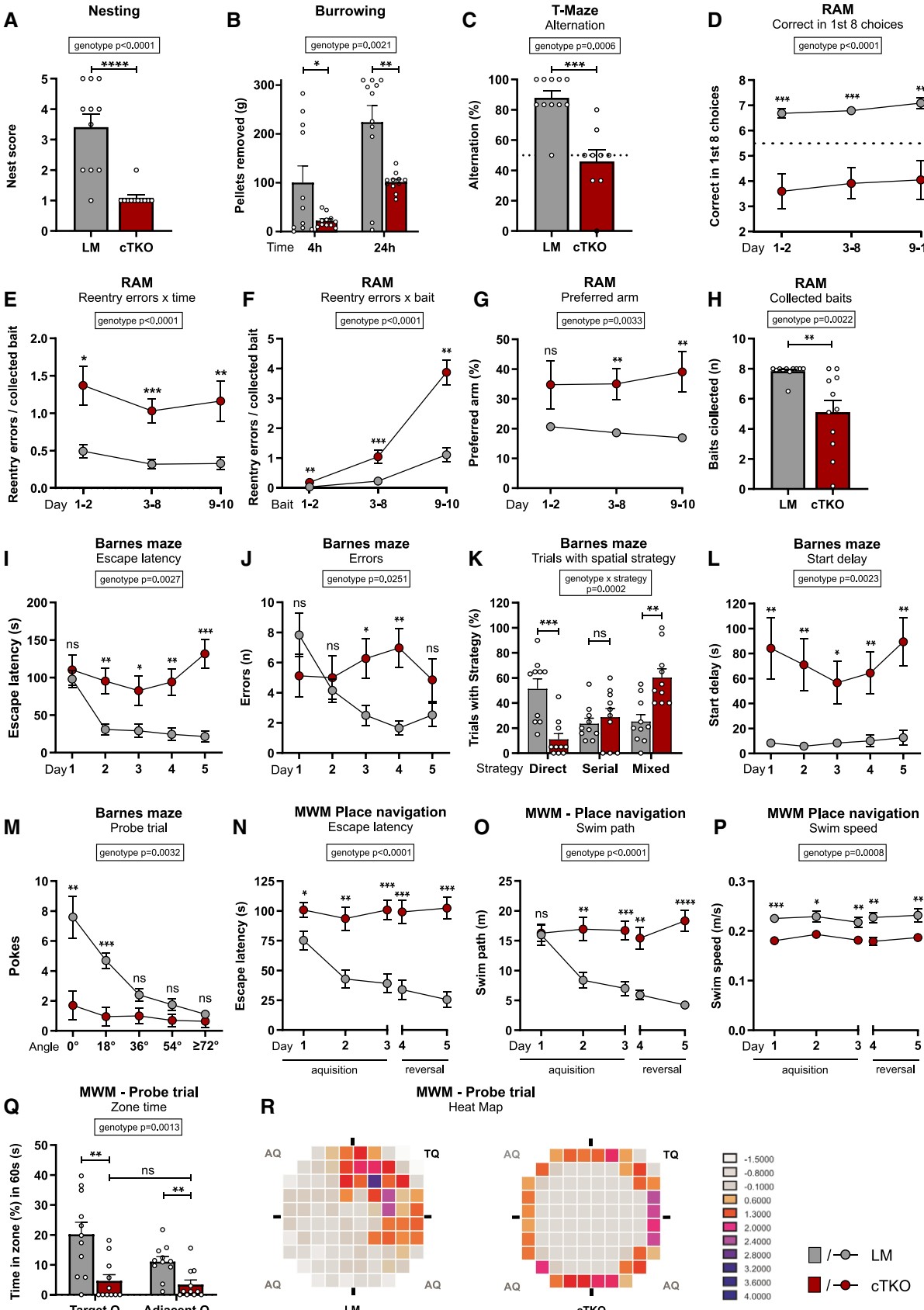

**Figure 6.**

**Figure 6. NexCre cTKOs show hyperactivity and severe deficits in learning and memory tasks.**

A NexCre cTKOs show a significantly reduced nest score [geno $F(1,18) = 26.00$, $P < 0.0001$].

B Burrowing: Pellets removed from tube after 4 and 24 h. Maximum to be removed = 310 g. Burrowing behavior was impaired in NexCre cTKOs [geno $F(1,18) = 12.95$, $P = 0.0021$; time $F(1,18) = 29.46$, $P < 0.0001$; time × geno $F(1,18) = 1.407$ ns].

C T-maze alternation test shows that spontaneous alternation is absent in NexCre cTKOs [geno $F(1,16) = 18.29$, $P = 0.0006$]. Chance level is indicated by a dotted line.

D Radial arm maze (RAM). During training, NexCre cTKO mice made less correct among the first 8 choices than LM controls [geno $F(1,18) = 24.72$, $P < 0.0001$; day $F(2,36) = 0.662$ ns; day × geno $F(2,36) = 0.044$ ns].

E RAM, NexCre cTKO animals made significantly more re-entry errors per bait than controls during training [geno $F(1,16) = 28.42$, $P < 0.0001$; day $F(2,32) = 1.209$ ns; day × geno $F(2,32) = 0.094$ ns].

F RAM, the number of memory errors increased in all groups with the numbers of baits already collected. This increase in re-entry errors per bait is considerably stronger in NexCre cTKO animals [geno $F(1,16) = 35.56$, $P < 0.0001$; bait $F(2,32) = 85.05$, $P < 0.0001$; bait × geno $F(2,32) = 3.138$, $P = 0.0570$].

G RAM, throughout training, NexCre cTKOs have a stronger tendency than controls to repeatedly visit a preferred arm [geno $F(1,18) = 11.51$, $P = 0.0033$; day $F(2,36) = 0.012$ ns; day × geno $F(2,36) = 0.965$ ns].

H RAM, in contrast to LM controls, NexCre cTKO animals did not collect all baits during the test [geno $F(1,18) = 12.98$, $P = 0.0022$]

I Barnes maze (BM), throughout training, latency to escape the maze did not decrease in NexCre cTKO animals [geno $F(1,16) = 12.63$, $P = 0.0027$; day $F(4,64) = 7.320$, $P < 0.0001$; day × geno $F(4,64) = 5.796$, $P = 0.0005$].

J BM, NexCre cTKO animals failed to reduce their number of errors during training [geno $F(1,16) = 2.316$ ns; day $F(4,64) = 2.991$, $P = 0.0251$; day × geno $F(4,64) = 6.341$, $P = 0.0002$].

K BM, trials of NexCre cTKO mice with a direct spatial strategy were strongly reduced compared to LM controls [strategy $F(2,32) = 2,324$ ns; geno × strategy $F(2,32) = 11.53$, $P = 0.0002$].

L BM, NexCre cTKO mice showed a higher latency to leave the starting point in the center of the maze [geno $F(1,16) = 13.17$, $P = 0.0023$; day $F(4,64) = 1.354$ ns; day × geno $F(4,64) = 0.995$ ns].

M BM, during the probe trial, NexCre cTKO animals made fewer pokes and showed no evidence of spatial selectivity for the goal position [geno $F(1,16) = 12.00$, $P = 0.0032$; angle $F(4,64) = 16.14$, $P < 0.0001$; angle × geno $F(4,64) = 8.902$, $P < 0.0001$].

N Morris water maze (MWM), escape latency. During place navigation training, NexCre cTKO animals showed no evidence of learning [geno $F(1,18) = 30.53$, $P < 0.0001$; day $F(4,72) = 13.35$, $P < 0.0001$; day × geno $F(4,72) = 11.77$ $P < 0.0001$].

O MWM, swim path. In contrast to LM controls, NexCre cTKO animals did not shorten their swim path over trials [geno $F(1,18) = 25.67$, $P < 0.0001$; day $F(4,72) = 9.273$, $P < 0.0001$; day × geno $F(4,72) = 12.86$, $P < 0.0001$].

P MWM, swim speed of NexCre cTKO animals was lower than in LM controls [geno $F(1,18) = 16.23$, $P = 0.0008$; day $F(4,72) = 1.787$ ns; day × geno $F(4,72) = 0.615$ ns].

Q MWM, in contrast to LM controls, NexCre cTKO showed no preference for the trained target quadrant [geno $F(1,18) = 14.57$, $P = 0.0013$; day $F(1,18) = 5.997$, $P = 0.0248$; day × geno $F(1,18) = 3.509$, $P = 0.0774$].

R MWM, representative occupancy plots during the probe trial. Dwell times were summed across all tested LM and cTKO mice, respectively, and then z-transformed. LM mice show a highly focused spatial search strategy for the trained platform position, while cTKO spend most of their time close to the wall.

Data information: (A,B,D,G,H,N-R): $n = 11$ animals/genotype; (C,E,F): $n = 11$ LM, $n = 9$ NexCre cTKO animals; (I-M): $n = 10$ animals/genotype. Balanced sex for all groups.
Data were analyzed using a mixed ANOVA model and are represented as mean ± SEM. *$P < 0.05$, **$P < 0.01$, ***$P < 0.001$, ****$P < 0.0001$, and ns not significant.

---

separation can be classified either as complex call types that are composed of several sounds at different frequencies (two syllable, frequency steps, harmonics, and waves) or contain more than one frequency change in a sound (composite), or as simple call types that are composed of single waves (chevre, short, downward, flat, and upward; see Fig EV4 for examples of sonograms; Scattoni *et al*, 2008; Takahashi *et al*, 2016). Overall, NexCre cTKO showed significantly more direct repetitions of calls of a specific type (Fig 5F). Interestingly, NexCre cTKO pups emitted significantly fewer complex calls including harmonics, waves, and composite calls, while the number of simple downwards calls was increased (Fig 5G). Thus, overall call type complexity was reduced in NexCre cTKOs (74.3 ± 3.4% simple versus 25.7 ± 3.4% complex calls) compared to LM controls (58.8 ± 3.6% simple versus 41.2 ± 3.6% complex calls; see Fig 5H).

Next, we studied social interactions of adult mice (age: 2 months) using the 3 chambers test (Fig 5I and J). In this test, the mouse can move freely between three equally sized interconnected chambers. When given the choice to spend time either with an unfamiliar mouse (stranger 1, same age and sex as tested mouse) present in a wire cage on one side of the central chamber or with an empty wire cage on the other side of the central chamber, NexCre cTKO mice failed to show a preference for the unknown mouse as compared to the novel object (Fig 5I), indicating impaired sociability in NexCre cTKO mice. In contrast, LM mice spend significantly more time exploring the stranger mouse (Fig 5I). In a second test

for preference of social novelty, mice had the choice between the now-familiar stranger 1 mouse and a second unfamiliar mouse (stranger 2, same age and sex). Unlike LM controls, that showed a clear preference for social novelty, exploration time of NexCre cTKO mice was not significantly different for stranger mouse 1 and 2 (Fig 5J). To exclude deficits in olfaction, that can impair social interactions, mice were tested for their ability to find buried food pellets (Fig EV5A). No significant difference was, however, detectable between genotypes. Further, as increased anxiety may impair sociability mice also underwent testing in the elevated plus maze paradigm (Fig EV5B and C). Mice of both genotypes spend similar time in the closed arms (Fig EV5B), indicating that impaired sociability is not due to increased anxiety in NexCre cTKO mice. While LM mice showed a very pronounced preference for the closed arms of the EPM, the behavior of cTKOs was considerably more variable (Fig EV5B), pointing toward a possible anxiolytic phenotype. Despite this, cTKOs still spend, on average, significantly more time in the closed arms, to an extent that was statistically comparable to LM controls. This more variable behavior was also reflected when analyzing the number of visits into open versus closed arms (Fig EV5C). cTKO mice visited open and closed arms with similar frequency, although the mean increase in the number of visits to open arms was largely due to 4 animals. This apparent increase in risk-taking behavior observed for some of the cTKOs contrasts, however, with their aversion to novelty, as seen by increased freezing in the start zone of the Barnes maze (Fig 6L).

In summary, NexCre cTKO revealed core phenotypes of ASD-like behavior including an increased frequency of repetitive stereotypic behaviors, impaired communication (USVs) in juvenile pups, impaired sociability and lack of preference for social novelty in adult mice.

## Discussion

Here, we report that lack of the entire APP family in excitatory forebrain neurons from E11.5 onwards impairs brain function at multiple levels: (i) NexCre cTKO mice show deficits in hippocampal lamination and agenesis of the corpus callosum; (ii) electrophysiological recordings in the hippocampus of NexCre cTKO mice revealed impaired basal synaptic transmission and severely reduced LTP, which was associated with impaired neuronal morphology and reduced spine density of hippocampal neurons; (iii) at the behavioral level, NexCre cTKO mice were not only severely impaired in several tests for learning and memory, but also exhibited core autism-like behaviors including stereotypic repetitive behaviors, reduced social communication, and impaired social interaction. Together, our study identifies essential functions of the APP family during brain development and for the function of networks mediating learning and social behavior.

### Impaired brain morphology

APP family members are widely expressed in excitatory and inhibitory neurons of the adult brain. In the developing telencephalon, APP and APLP2 are expressed in the proliferative zones (the ventricular zone and the subventricular zone), while APLP1 is expressed in the cortical plate suggesting a possible role of the APP family in brain development (Lopez-Sanchez *et al*, 2005). Using stereology, however, we observed no deficits in cortical volume of either young adult (5–6 months) or aged (18–20 months) NexCre cTKO mice, indicating that APP and the APLPs are not essential for the genesis and survival of cortical neurons *in vivo*. This is consistent with findings from a recent study, in which APP family proteins were deleted postnatally using the CamKII-Cre driver line (Lee *et al*, 2020). We failed to detect gross abnormalities in cortical architecture, while earlier studies using *in utero* RNAi knockdown (Young-Pearse *et al*, 2007; Shariati *et al*, 2013) or constitutive germline TKO mice (Herms *et al*, 2004) had revealed abnormal focal neuronal positioning. These differences are likely due to the pattern (cell types and developmental time point) of Cre expression that is restricted in NexCre cTKO to postmitotic excitatory neurons, while in germline TKO mice, gene deletion will affect all cell types including proliferating progenitor cells and glia. Despite grossly normal cortical layering, we observed striking abnormalities in the hippocampus with a bilaminar appearance of the CA1 band that was split into a superficial and deep layer. Less pronounced alterations were also evident in the CA3 region. While the underlying mechanism deserves further investigation, it is interesting that a similar phenotype (also lacking cortical defects) has been observed for mouse mutants of genes involved in regulating microtubuli including Lis, Dcx, and Dclk1/Dclk2 (reviewed in Stouffer *et al*, 2016). This suggests that APP and APLPs may modulate cytoskeleton dynamics of migrating neurons and/or the adhesion of neuronal precursors with radial glia

cells. We consider the latter more likely, as in a previous study no differences in the speed of migration nor in the morphology of cortical TKO progenitors were detectable (Shariati *et al*, 2013). Moreover, during later stages of hippocampal development neurons display a climbing mode of migration, in which they migrate in a zig-zag fashion between multiple radial glia fibers. This feature, that is specific for hippocampus, has been suggested to contribute to the increased vulnerability of the hippocampus to malformations (Belvindrah *et al*, 2014). As APP mediates cell–substrate and cell–cell interactions (Soba *et al*, 2005; Müller *et al*, 2017), roles for APP in contact guidance have been suggested (Rama *et al*, 2012; Sosa *et al*, 2013; Wang *et al*, 2017). Despite this, axonal connectivity was only mildly impaired in APP-KO single mutants that exhibit smaller forebrain commissures (Magara *et al*, 1999) and rather subtle deficits in the retinotectal system (Osterhout *et al*, 2015; Marik *et al*, 2016). Here, we found a very high incidence of callosal agenesis or severe callosal dysgenesis in NexCre cTKO mice, whereas this phenotype was absent in previously generated APLP single mutants or NexCre conditional double mutant mice (cDKO) lacking APP and APLP2 (Heber *et al*, 2000; Hick *et al*, 2015). This indicates an important so far unrecognized role of APLP1 for axonal wiring and circuit formation *in vivo*, likely overlapping with that of APP and APLP2. Although disturbed brain development frequently manifests with intellectual disability and/or ASD in humans, we hypothesize that the hippocampal lamination deficit and the callosal agenesis may contribute to, but is unlikely to be the major cause of behavioral impairments. In this regard, Dcx (-/Y) mice, that exhibit similarly impaired hippocampal lamination and callosal agenesis, exhibit normal hippocampus-dependent learning and memory (Kappeler *et al*, 2007; Germain *et al*, 2013).

### Lack of APP family disrupts learning and leads to ASD-like behavior

The lack of all APP family members during development disrupts behavior in several domains. NexCre cTKOs show pronounced hyperactivity in the familiar home cage with massively overshooting dark phase activity. Also in the open field, which assesses locomotor activity in a novel environment, NexCre cTKOs failed to habituate and showed a paradoxical increase of activity over time. Activity was, however, not explorative but consisted mostly of stereotypic running along the wall with excessive avoidance of the center field. Both lack of habituation and severely impaired nesting and burrowing are consistent with, but not limited to, hippocampal dysfunction (Deacon & Rawlins, 2005; Leussis & Bolivar, 2006).

While muscle strength, motor learning, and coordination were normal, NexCre cTKOs performed extremely poorly in all cognitive tests. To assess spatial learning and memory, mice underwent testing in the T-maze, 8-arm radial maze, Barnes maze, and Morris water maze. In all of these tests, the performance of NexCre cTKO mice was strongly impaired with no evidence of learning. Spontaneous alternation in the T-maze was completely disrupted indicating impaired working memory. In the water maze, the behavior of NexCre cTKOs was dominated by passive floating and persistent wall-hugging, which was also observed in cued navigation. Here, NexCre cTKOs also revealed increased impulsivity and frequently jumped off the cued platform. In dry mazes, such as the Barnes maze, mutants often showed initial freezing with delayed onset of

locomotion suggesting aversion to novelty. Overall, the performance of NexCre cTKOs in the Barnes maze was very poor and NexCre cTKOs failed to develop a spatial search strategy. Notably, due to the high incidence of perseverative behaviors in NexCre cTKOs, the deficits in learning and memory cannot be solely attributed to impaired cognition. Our data rather suggest that the inability of NexCre cTKOs to suppress their genuine hyperactivity and/or their lack of motivation/attention constitutes major factors contributing to their poor performance. As such, hyperactivity, impulsivity, perseverative behavior, and behavioral inflexibility of NexCre cTKO mice are reminiscent of attention deficit hyperactivity disorder that is frequently observed as a comorbidity in ASD patients (Taurines *et al,* 2012; Lai *et al,* 2019). Moreover, aversion to novelty and behavioral inflexibility are also key features of compromised executive functions that are impaired in patients with frontal lobe damage, aging-associated cognitive decline, and dementia including AD (Miller, 2000; Guarino *et al,* 2018).

In this study, we demonstrated that the lack of the APP family in NexCre cTKO mice does not only disrupt learning but also induces core autism-like deficits (APA, 2013). This underscores the importance to include developmental aspects when assessing APP family functions and distinguishes our study from other approaches, which investigated functions in the adult forebrain (Lee *et al,* 2020). In addition to stereotypic repetitive rearing and climbing, NexCre cTKO mice showed impaired communication. NexCre cTKO pups emitted fewer calls of shorter duration and reduced complexity. In the three chambers test, adult NexCre cTKO mice showed not only impaired sociability but also lacked a clear preference for social novelty. Despite these ASD-like abnormalities of NexCre cTKO mice, APP or the APLPs have so far not been identified as risk genes in genetic screens for ASD (Satterstrom *et al,* 2020). This is, however, not too surprising given that the ASD-like phenotype only emerges upon inactivation of all three family members during development. This points again toward a crucial, overlapping role of APLP1 for normal brain physiology and establishes an essential role of the APP family for brain circuits important for cognition and social behavior. Nevertheless, and suggestive of a possibly more prominent role of APP for human brain physiology, an infant with a homozygous truncating mutation in the APP N-terminus was reported to show developmental delay, microcephaly, callosal dysgenesis, and seizures (Klein *et al,* 2016). As mice with an adult knockout of the APP family did not show ASD-like traits and lacked both the deficits in spine density and the severe disruption of learning (Lee *et al,* 2020), the behavioral phenotype of NexCre cTKO likely represents a neurodevelopmental disorder. In addition to the earlier time point of gene deletion, the complete and constitutive inactivation of APLP1 may further contribute to the severe impairments of NexCre cTKO mice, as CamKII-Cre driven postnatal cTKO mice showed incomplete gene inactivation with residual APLP1 expression at about 50 % of wild-type level (Lee *et al,* 2020).

### Disrupted learning and ASD-like behavior are associated with impaired synaptic plasticity

The severe behavioral phenotype of NexCre cTKO mice is associated with impairments at multiple levels and in several brain regions. CA1 neurons of NexCre cTKO mice exhibit altered neuronal morphology with a reduction in the total dendritic length

of the apical dendrite of sCA1 cells and fewer primary dendrites in both superficial and deep CA1 cells. Consistent with impaired LTP and a role for transsynaptic adhesion mediated by APP family proteins (Soba *et al,* 2005; Schilling *et al,* 2017), lack of the APP family also impaired synapse density in the hippocampus. Interestingly, CA1 cells were differentially affected along the radial axis, with spine density deficits confined either to midapical dendrites of sCA1 cells or to basal dendrites of dCA1 cells. This is in line with differential connectivity of superficial and deep CA1 cells. Regarding the Schaffer collateral projections, proximal CA3 cells (located close to the DG) project primarily to superficial portions of the stratum radiatum, while distal CA3 cells (located more closely to CA1) project predominantly to stratum oriens (Ishizuka *et al,* 1990) (reviewed in Soltesz & Losonczy, 2018). Moreover, distal CA3/CA2 cells have recently been shown to innervate more strongly dCA1 cells (Kohara *et al,* 2014). Together, our data identify, to our knowledge for the first time, a selective vulnerability of deep versus superficial CA1 cells for spine density deficits in mice with a genetically induced disease-like phenotype and thus extend the diverse properties of CA1 cell types along the radial axis (Slomianka *et al,* 2011; Geiller *et al,* 2017; Soltesz & Losonczy, 2018). It will also be interesting to investigate whether cTKO mice show possible differences in the relative abundance of mature versus immature spines and their turnover, as APP-KO mice showed impairments in environmental enrichment-induced spine plasticity within the cortex (Zou *et al,* 2016).

LTP is a process whereby brief periods of synaptic activity, like TBS used here, can produce a long-lasting increase in synapse strength. There are three properties of LTP: (i) cooperativity, (ii) input specificity, and (iii) associativity, which are essential for learning and memory in mammals (Kandel *et al,* 2013). Associativity means that co-activation of weak inputs with strong inputs onto the same neuron can strengthen the weak input. In our case, NMDAR function is not grossly altered, but LTP is compromised and the number of inputs to a CA1 pyramidal neuron is reduced, which most likely reduces the power of associativity and cooperativity. Without altering the functional properties of a single synapse, this would already lower the probability to induce LTP and is the most likely explanation for the LTP deficits observed in cTKO mice. Together, the dramatic deficits in LTP induction and maintenance, as well as in basal synaptic transmission between CA3 and CA1 neurons, indicate a dysfunctional hippocampal circuitry that is likely to result in the behavioral impairments of NexCre cTKO mice.

Interestingly, reduced synapse density and impaired synaptic plasticity have previously been associated not only with deficits in learning and memory, but also with ASD-like traits in a wide range of mouse mutants of synaptic proteins, notably in other mutants of synaptic adhesion proteins including neuroligin/neurexin and LRRC4 (Baig *et al,* 2017; Um *et al,* 2018) (reviewed in Lin *et al,* 2016). Despite the absence of major structural abnormalities of the cortex, the complex behavioral phenotype suggests that also cortical network integrity might be affected, which needs more detailed further studies. In this regard, previous studies of constitutive APP-KO mice revealed abnormalities of the cortical oscillatory network including impaired cross-regional coupling between hippocampus and prefrontal cortex (Zhang *et al,* 2016).

Since the molecular cloning of APP more than 30 years ago (Goldgaber *et al,* 1987; Kang *et al,* 1987; Tanzi *et al,* 1987) and the

subsequent identification of autosomal dominant APP mutations linked to familial forms of AD (Goate *et al*, 1991), the intensely studied role of APP and Aβ for AD etiology had occluded the analysis of its physiological functions (Müller *et al*, 2017). The findings of this study highlight the essential role of the APP family for synaptogenesis, synaptic plasticity, and a wide range of behaviors including learning and social interactions. Thus, our findings reinforce the need for intensifying research directed at the physiological role of APP and the frequently neglected APLPs to complement our understanding of AD as a disease of synaptic and circuit dysfunction.

# Materials and Methods

### Mice

Experiments involving animals were performed in accordance with the guidelines and regulations set forth by the German Animal Welfare Act and the Regierungspräsidium Karlsruhe, Germany. Animals were housed in the same room with a 12-h/12-h light/dark cycle in Makrolon Type II (360 cm$^2$) cages with standard bedding, either alone or in groups, and had *ad libitum* access to standard chow and water. Generation and genotyping of individual mice were described previously: APP$^{flox}$ and APLP2$^{flox}$ (Mallm *et al*, 2010); APLP1-KO (Heber *et al*, 2000); and NexCre (Goebbels *et al*, 2006). The mutant mouse lines with modifications of only one family member (APPflox, APLP2flox, APLP1-KO) had previously been backcrossed to C56BL6 for more than six generations. The three strains were crossbred until the desired genotypes were generated. Final matings were APP$^{flox/flox}$/APLP2$^{flox/flox}$/APLP1$^{-/-}$ x APP$^{flox/flox}$/ APLP2$^{flox/flox}$/APLP1$^{-/-}$ NexCre$^{+/T}$ to obtain 50% NexCre cTKOs and 50% APLP1-KO littermate (LM) controls (see mating scheme depicted in Fig EV1B).

### Stereology

Cell number and volume estimates were performed in every 10$^{th}$ 20-μm-thick coronal Giemsa-stained section of methacrylate-embedded hemispheres. The Cavalieri estimator, using a 200 × 200 μm point grid, was used to estimate volume. To facilitate comparisons between animals and groups, volumes were distributed into 12 equidistant bins along the rostro-caudal axis of each animal (Amrein *et al*, 2015). The optical fractionator was used to estimate cell number (West *et al*, 1991) using 25 × 25 × 10 μm disector probes spaced 130 μm apart along the x- and y-axes and a 2 μm top guard zone. Volume and cell number estimate precisions were assessed using the Gundersen–Jensen *CE* (coefficient of error) estimator with a conservative m = 0 (Gundersen *et al*, 1999; Slomianka & West, 2005), which provides an estimate of the CE that can be expected from replications of the volume and cell number measurements.

### Immunohistochemistry

Animals were killed with $CO_2$ and transcardially perfused with PBS, followed by a 5-min fixation with 4% PFA in PBS. The brain was dissected from the skull and postfixed overnight at 4°C in 4% PFA in PBS. 40-μm coronal or horizontal sections were cut on a Vibratome (HM650V microtome) and collected in cold PBS in 24-well plates. Slices were blocked and permeabilized (5% BSA (w/v), 5% NGS (v/v), 0.4% Triton X-100 (w/v) in PBS) overnight at 4°C and stained free-floating in 300 μl of the appropriate dilutions of primary antibody (in PBS + 5% NGS + 0.2% Triton X-100) again overnight at 4°C. After washing with PBS, 300 μl of the corresponding secondary antibody dilutions (in PBS + 0.1% BSA + 0.05% Triton X-100) were applied and incubated for 2 h at RT. After washing, the sections were stained with PBS-DAPI, collected on Superfrost microscope slides (Menzel), and mounted in Mowiol. Antibodies were as follows: calbindin (CB; rabbit, 1:3,000, #CB38, swant, Switzerland); Ctip2 (rat, 1:200, #25B6, Abcam, United Kingdom); GFAP (rabbit, 1:3,000, #173002, Synaptic Systems, Germany); Iba1 (rabbit, 1:500, #234003, Synaptic Systems, Germany); goat anti-rabbit Cy3 (1:1,500, #711-165-152, Jackson ImmunoResearch Laboratories, USA); goat anti-mouse Cy5 (1:500, #A10524, Thermo Fisher Scientific, USA); goat anti-rat Alexa 488 (1:500, #112-545-167, Jackson ImmunoResearch, United Kingdom); and goat anti-guinea pig Alexa 488 (1:500, #106-545-003, Jackson ImmunoResearch, United Kingdom). Images were taken with an Axio Observer Z1 (Zeiss, Germany) and a Leica TCS SP5II (Leica, Germany).

### Neuronal morphology and spine density analysis

Superficial and deep CA1 pyramidal neurons used for morphological analysis were filled with a solution containing 0.1–0.5% biocytin (Sigma-Aldrich) through the patch pipette while recording. Acute slices were fixed in 4% Histofix (Carl Roth) after recording. After 2–10 days, the slices were washed in 1× PBS (phosphate-buffered saline) for 3× 10 min. Permeabilization was performed for 1 h in 0.2% PBST (0.2% Triton X-100 in 1× PBS). Slices were stained overnight with Alexa 594-conjugated Streptavidin directed against biocytin (Life Technologies). On the next day, the slices were washed again for 3× 10 min in 1× PBS. After air-drying the slices at RT for 1 h, they were mounted with a coverslip in ProLong Gold Antifade (Life Technologies).

#### *Image acquisition*

Images of filled neurons were acquired at the inverted fluorescence microscope Axio Observer Z1 using Plan Apo 20x/0.8 DICII and Plan Apo 63x/1.4 Oil DICII objectives (Zeiss). Overview images of the whole neuron for reconstruction were taken with a 20x objective and a z-step size of 0.5 μm. Basal and apical dendrites were imaged individually with two overlapping stacks. More detailed images of basal and apical dendritic segments for spine density analysis were acquired with a 63× oil objective and a z-step size of 130 nm. Exposure time was individually set for each cell so that the complete range of the grayscale was used.

#### *Neuronal morphology and spine counts*

Biocytin-filled hippocampal CA1 neurons were manually reconstructed using the Neurolucida software (MicroBrightField) by an experimenter blind to genotype. Neurons were only included in Sholl analysis if they showed a completely filled apical or basal tree and well-defined dendritic endings. The morphometric Sholl analysis was done using the NeuroExplorer software (MicroBrightField). In short, a series of concentric spheres (centered on the soma) was drawn with an intersection interval of 30 μm and the number of dendrite crossing each sphere as well as the dendritic length in

between each sphere was calculated. This analysis was done separately for basal and apical dendrites of CA1 pyramidal cells and was plotted against the distance from the soma.

For evaluation of basal dendritic spine density, at least three different dendritic segments of the basal dendritic arbor were imaged. They had to fulfill the following criteria: (i) lie mostly horizontally to the slice surface, (ii) be at least 20 μm away from the soma, and (iii) have a comparable thickness. The minimum basal dendritic length imaged per neuron was 100 μm. For evaluation of midapical dendritic spine density, at least 3 different dendritic segments of the apical tree were imaged. Midapical was defined as the middle third of the length of the apical dendrite measured from the origin of the apical dendrite from the soma to the endpoint of the tufts. Dendritic segments used for evaluation had to fulfill the following criteria: (i) be of second or third order to assure comparable shaft thickness, (ii) lie in the middle third of the main apical dendrite and (iii) be longer than 10 μm. The minimum midapical dendritic length imaged per neuron was 100 μm. Files in the zvi format were imported into ImageJ (NIH) using the BioFormats Importer. After adjusting, images were saved in the TIFF format. Dendritic spines were manually counted using the Neurolucida and NeuroExplorer software (MicroBrightField) following the criteria of Holtmaat (Holtmaat *et al*, 2009) with minor modifications: (i) All spines that protruded laterally from the dendritic shaft and exceeded a length of 0.4 μm were counted. (iii Spines that protruded into the z-plane were only counted if they exceeded the dendritic shaft more than 0.4 μm to the lateral side. (iii) Spines that bisected were counted as two spines. (iv) Spines had to be at least 10 μm away from branching points and the soma. Spine density was expressed as spines per μm of dendrite. Prior to statistical analysis and blind to genotype, neurons were excluded if the image quality (poor signal to noise ratio) was not sufficient for counting of spines. Data acquisition and analysis were performed blind to genotype.

## Electrophysiology

### Extracellular field recordings

*In vitro* extracellular recordings were performed on acute hippocampal slices of 4- to 5-month-old NexCre cTKOs or LM controls ($N$ = 5 animals/genotype). In between, animals were group-housed in a temperature- and humidity-controlled room with a 12-h light–dark cycle and had access to food and water *ad libitum*.

### Slice preparation

Acute hippocampal transversal slices were prepared from isoflurane-anesthetized individuals. Following decapitation, the brain was removed and quickly transferred into ice-cold carbogenated (95% $O_2$, 5% $CO_2$) artificial cerebrospinal fluid (ACSF) containing 125.0 mM NaCl, 2.0 mM KCl, 1.25 mM $NaH_2PO_4$, 2.0 mM $MgCl_2$, 26.0 mM $NaHCO_3$, 2.0 mM $CaCl_2$, and 25.0 mM glucose. The hippocampus was sectioned into 400-μm-thick transversal slices with a vibrating microtome (VT1200S, Leica) and maintained in carbogenated ACSF at room temperature for at least 1.5 h. Slices were placed in a submerged recording chamber and perfused with carbogenated ACSF (32°C) at a rate of ~1.5 ml/min. Field excitatory postsynaptic potentials (fEPSPs) were recorded in stratum radiatum of CA1 region with a borosilicate glass micropipette (resistance 2–4 MΩ) filled with 3 M NaCl at a depth of ~150–200 μm.

Monopolar tungsten electrodes were used for stimulating the Schaffer collaterals at a frequency of 0.1 Hz. Stimulation intensity was adjusted to ~40% of maximum fEPSP slope for 20 min baseline recording. LTP was induced by applying theta burst stimulation (TBS: 10 trains of four pulses at 100 Hz in a 200-ms interval, repeated three times).

Basal synaptic transmission properties were analyzed via input–output (IO) measurements, and short-term plasticity was examined by performing paired-pulse stimulation experiments. The IO measurements were performed either by application of defined current values (25–250 μA) or by adjusting the stimulus intensity to certain fiber volley (FV) amplitudes (0.1–0.8 mV). Presynaptic function and short-term plasticity were assessed with the paired-pulse facilitation (PPF) paradigm by applying a pair of two closely spaced stimuli in inter-stimulus intervals (ISIs) ranging from 10 to 160 ms.

### Data analysis and statistics of extracellular field recordings

Data of electrophysiological recordings were collected, stored, and analyzed with LABVIEW software (National Instruments). The initial slope of fEPSPs elicited by stimulation of the Schaffer collaterals was measured over time, normalized to baseline, and plotted as average ± SEM. Analysis of the PPF data was performed by calculating the ratio of the slope of the second fEPSP divided by the slope of the first one and multiplied by 100. The statistical analysis was performed using Microsoft Excel or GraphPad Prism (GraphPad). Data obtained between two genotypes or two different experimental conditions were compared using an unpaired two-tailed Student's *t*-test. All data are indicated as mean ± SEM. Values of $P$ < 0.05 were considered significant and plotted as follows: *$P$ < 0.05, **$P$ < 0.01, and ***$P$ < 0.001. Data acquisition and analysis were performed blind to genotype.

### Patch-clamp experiments

NexCre cTKO mice and LM controls (age: 4 months) were used to obtain 250-μm-thick acute transversal brain slices of the hippocampus. Briefly, mice were anesthetized with isoflurane and intracardially perfused with sucrose dissection (SD) solution containing 212 mM sucrose, 3.0 mM KCl, 1.25 mM $NaH_2PO_4$, 7.0 mM $MgCl_2$, 26.0 mM $NaHCO_3$, 0.02 mM $CaCl_2$, and 10.0 mM glucose. Brains were quickly removed and placed into ice-cold SD solution in the holding chamber of a tissue slicer (Leica, Germany; razor blade: Personna, USA). Immediately after preparation of the slices, they were incubated at 37°C for 15 min in ACSF containing 125 mM NaCl, 2.5 mM KCl, 2 mM $CaCl_2$, 1 mM $MgCl_2$, 25 mM $NaHCO_3$, 1.25 mM $NaH_2PO_4$, and 25 mM glucose, bubbled with 95% $O_2$/ 5% $CO_2$ to pH 7.4. The holding chamber was slowly cooled down to RT, and slices were incubated for 45 min before recordings. Miniature excitatory postsynaptic currents (mEPSCs) and inhibitory postsynaptic currents (mIPSCs) were recorded using glass capillaries filled with an internal solution containing 120 mM Cs-methane sulfonate, 8 mM NaCl, 10 mM CsCl, 0.3 mM GTP, 2 mM MgATP, 10 mM $Na_2$-phosphocreatine, 0.2 mM EGTA, and 10 mM HEPES, pH 7.3. mEPSCs were recorded in ACSF at RT (superficial layer) or 32°C (deep layer) in the presence of 10 μM SR95531 hydrobromide (gabazine; Biotrend, Germany), 50 μM APV (Biotrend, Germany), and 1 μM TTX (Biotrend, Germany). mIPSCs were recorded in 32°C ACSF in the presence of 10 μM CNQX (Biotrend, Germany), 10μM APV (Biotrend, Germany), and 1 μM TTX. mEPSCs and mIPSCs

were recorded in voltage clamp at a holding potential of −70 mV. Series resistance was monitored but no compensation was performed. Miniature events were detected using Clampfit software (Molecular Devices, USA) with a template search. Statistical analyses were performed with GraphPad Prism (GraphPad software USA) using Mann–Whitney tests. Data are presented as mean ± SEM.

For the analysis of the ratio of AMPA to NMDA receptor-mediated currents (A/N), recordings were performed in ACSF (at RT) in the presence of 10 μM SR95531 hydrobromide (gabazine; Biotrend, Germany). Extrasynaptic stimulation was applied via a chlorinated silver wire located inside a borosilicate glass capillary filled with ACSF. The stimulus was generated by a stimulus isolator (WPI, USA) connected to an amplifier (HEKA, Germany) and triggered via the Patchmaster software (HEKA, Germany). Holding potentials for recording AMPA and NMDA receptor-mediated currents were −70 and +40 mV, respectively. The amplitude of NMDA receptor-mediated currents was measured 35 ms after the stimulus to avoid AMPA receptor-mediated current contamination. Data acquisition and analysis were performed blind to genotype.

## Behavioral analysis

### Neuromotor behavior and cognitive tests
#### Animals
22 mice were tested in total: 11 NexCre cTKO APP$^{flox/flox}$/APLP2$^{flox/flox}$/APLP1$^{−/−}$NexCre$^{+/T}$; five females, six males) and 11 littermate (LM) controls (APP$^{flox/flox}$/APLP2$^{flox/flox}$/APLP1$^{−/−}$; five females, six males). Animals were housed under a 12/12-h light–dark cycle (lights on at 20:00) in groups of 2–5, unless individual housing was required by experimental protocols or to prevent fighting. Testing occurred during the dark phase under dim light (approximately 22 lux) and if not stated otherwise, identity of genotype was blinded to the experimenter. Mice were transferred to the testing room 30 min before testing. Procedures were approved by the Veterinary Office of the Canton of Zurich (license ZH044/15, #26394).

Animals were aged 5 months at the beginning of behavioral testing. Test sequence was as follows: open field, grip test, rotarod, water-maze place navigation, burrowing, nesting, T-maze, radial maze, Barnes maze, and water-maze cue navigation (see Appendix). Tests including recovery periods in between lasted 7 weeks. T-maze: Two cTKOs showed immobility in the starting arm and were excluded from the analysis as they performed only very few choices. One female LM did not perform all tests as it was found dead after 6 weeks of testing. From one female NexCre cTKO, we recorded only incomplete Barnes maze data due to seizures. This animal was therefore excluded from the Barnes maze analysis. None of the other animals had to be excluded during the tests.

### Open field
Activity was tested as described previously (Madani *et al*, 2003). In brief, animals were tested on two consecutive days for 10 min in the open field, a circular arena of 150 cm in diameter. Mice were tracked using Noldus EthoVision 11.5 software (www.noldus.com). For analysis of movement patterns, the arena was divided into a wall zone (18% of surface, 7 cm wide), a center zone (50%), and a transition zone in between. To analyze stereotypic or repetitive behavioral patterns, the arena was sub-divided into 5 × 5 (numbered) quadratic tiles, and movement paths of the animals were analyzed for repeating sequences in moving from one specific tile to the next. Stereotypy index is expressed as % of the number of tiles that were crossed in a repeated sequence over total tile crossings.

### Motor behavior, grip strength
Forepaw grip strength was measured as described previously (Ring *et al*, 2007) using a newton meter (max. force: 300 g). Animals had to hold on a metallic bar (4 cm long, 2.5 mm in diameter) attached to the horizontally positioned newton meter. Mice were held by the tail and allowed to grasp the bar with both forepaws. They were then gently pulled back until they released the bar. Mice were tested on two consecutive days for five trials each. For analysis, values of maximal pulling force were averaged.

### Motor behavior, RotaRod
The RotaRod (Ugo Basile, model 47600, Comerio, Italy) consisted of a rotating drum with a minimum speed of 2 round per minute (rpm) to maximal 40 rpm. Rotation speed was increased linearly until maximum speed was reached after 290s. Animals were tested for 5 sessions on the same day, and each session was terminated once the animal fell down the drum or after 300 s the latest. Time at which animals for the first time clung to the drum (full circle ride) and time and acceleration at which the animals dropped off the drum were evaluated. For analysis, values were averaged.

### Mouse-specific behavior, Burrowing
The burrowing test was done as described previously (Deacon, 2006b). In short, a gray plastic tube was filled with 310 g standard diet food pellets and placed at a slight angle into a type III standard mouse cage equipped with normal bedding, a mouse shelter, and water ad libitum. The lower end of the tube was closed, resting on the cage floor. The open end was supported 3.5 cm above the floor by two metal bolts. At the beginning of the dark period, mice were placed individually in the test cages, which were placed in their familiar animal room. At 4 h, and again at 24 h after experimental start, the amount of non-displaced food (food still in the tube) was weighted. Consumed food by the animals (2 ± 0.5 g) was a very small proportion of the 310 g available and approximately equal across groups.

### Mouse-specific behavior, Nesting test
Nest building was studied as described (Deacon, 2006a). At the beginning of the dark phase, mice were placed in individual testing cages (type II) in their familiar animal room containing regular bedding and a Nestlet of 3 g compressed cotton (Ancare, Bellmore, USA). After 24 h, the nest-building activity of the mice was assessed on a rating scale of 1 to 5: 1 = Nestlet > 90% intact, 2 = Nestlet 50–90% intact, 3 = Nestlet mostly shredded but no identifiable nest site, 4 = identifiable but flat nest, and 5 = crater-shaped nest. Remaining intact parts of the Nestlet were weighted.

### Water-maze place navigation
Place navigation was assessed as described previously (Ring *et al*, 2007). A white circular pool (150 cm diameter) contained milky water (24–26°C). Acquisition training consisted of 18 trials (6 per day, inter-trial interval 30–60 min) during which the submerged platform (14 × 14 cm) was left in the same position. Trials lasted a maximum of 120 s. To monitor reversal learning, the platform was moved to the opposite position for 2 additional days of testing (six

trials per day). Trials were video-tracked using a Noldus EthoVision. Raw data were transferred to the software Wintrack (www.dpwolfe r.ch/wintrack) for analysis. Results were plotted in bins of three trials. Passive floating episodes were defined as immobility or decelerations with speed minimum < 0.06 m/s and removed from the data before calculating swim speed. A slightly modified version of Whishaw's error was calculated as path (%) outside a 18.5 cm wide corridor connecting release point and goal. Cumulative search error was determined by summing the distances to target measured at 1-s intervals and subtracting value that would be obtained for an ideal direct swim. Finally, wall-hugging was quantified by time (%) spent in a 10 cm wide wall zone. The first 30 s of the reversal trial served as probe trial to test for spatial retention.

### Water-maze cue navigation

On two consecutive days, mice were tested with the cued variant of the Morris water maze. For this, the location of the platform was marked with a black-and-white striped inverted pyramid (height 11 cm, base of pyramid 11 × 11 cm) above the water. Animals were again tested in 6 trials per day, position of the flagged platform changed with each trial. Trials were video-tracked and analyzed as in place navigation.

### T-maze

Spontaneous alternation on the T-maze was assessed as described in Deacon & Rawlins, 2006. The T-maze was made of gray PVC. Each arm measured 30 × 10 cm. A removable central partition extended from the center of the back-goal wall of the T to 7 cm into the start arm. This prevented the mouse from seeing or smelling the non-chosen arm during the sample run, thus minimizing interfering stimuli. The entrance to each goal arm was fitted with a guillotine door. Each trial consisted of an information-gathering, sample run, followed immediately by a choice run. For the sample run, a mouse was placed in the start arm, facing away from the choice point with the central partition in place. The mouse was allowed to choose a goal arm and was confined there for 30 s by lowering the guillotine door. Then, the central partition was removed, the mouse replaced in the start arm, and the guillotine door was raised. Alternation was defined as entering the opposite arm to that entered on the sample trial (whole body, including tail). Three trials were run per day with an inter-trial interval of approximately 60 min. Each mouse received six trials in total, and for data analysis, the percentage of correct choices was calculated.

### Radial maze

The working memory procedure on the 8-arm radial maze was carried out as described previously (Weyer et al, 2011). Eight arms (7 × 38 cm) with clear Perspex tunnels (5 cm high) extended from an octagonal center platform (diameter 18.5 cm, distance platform center to end of arm 47 cm). The maze was placed 38 cm above the floor in a room rich in salient extramaze cues (same room as for open field testing). At the end of each arm, a metal cup (3 cm diameter) was lowered 1 cm to floor recess containing one millet seed as bait (total ca 0.05 g); thus, mice could not see the bait without completely entering the arm. Prior to the test, mice were gradually reduced to 85% of their free-feeding body weight for two days using a premeasured amount of chow, body weight was measured daily, and 85% body weight was maintained throughout the test period.

Water was available ad libitum. One day before test begin, mice were placed for 10 min into the baited radial maze for habituation. For testing, each mouse performed 1 session per day of maximally 10 min or until all eight seeds were collected. Test duration was 10 days. Mice were released in the center platform, performance of the animals was video-tracked, and first visits to each arm and consumption of seeds were recorded manually. Using the video-tracking information, we calculated the number of correct choices among the first eight, as well as the number of re-entry errors as a function of trial and of baits already collected. In addition, preferences for arm visits were analyzed. Error-free trials with one visit to each of the eight arms yielded preferred arm visits of 12.5%, corresponding to chance level without a preference for any arm. Re-entries into already visited arms would yield values between 12.5 and 25%. Scores higher than 25% indicated excessive entry into one particular, preferred arm.

### Barnes maze

The Barnes maze was made of a circular arena (1 m in diameter), placed 64 cm above ground. 20 holes (5 cm in diameter) were evenly distributed at the margin of the platform. A black escape-/goal box attached to the underside of a hole, equipped with a ramp inside, provided easy access to the dark escape. Tests were run in a brightly lit room for 5 days. Each day, animals were trained in four trials, 3 min each, with a fixed position of the escape box. On the last day, one additional trial without escape box (probe trial) was run for 3 min. For each trial, animals were placed under a circular opaque start box in the platform center for 30 s. Trial started with removing the start box. If the animal successfully escaped into the goal box, the start box was placed over the whole with the goal box for another 30 s to prevent re-emergence of the mouse. Mice that did not succeed in finding the escape box within the given time were gently guided to the escape box during the first day of testing. Trials were video-tracked. Tracking data were used to calculate start delay (trial start until exit of start area), escape latency (exit of start area until disappearance of the animal), and number of errors (nosepokes into incorrect holes until first poke into the correct hole). In addition, trials were categorized according to search strategy: direct (max 1 error with absolute deviation angle < 27°), serial (no center crosses and > 33% pokes to consecutive holes), or mixed (all remaining trials). During the probe trial, pokes were categorized according to deviation from the target hole.

### Statistics (neuromotor behavior and cognitive tests)

Data of the NexCre cTKO line were analyzed using an ANOVA model with genotype (NexCre cTKO and LM) and sex (F, M) as between subject factors. Within-subject factors were added to explore the dependence of genotype effects on place, time, or stimulus. Variables with strongly skewed distributions or strong correlations between variances and group means were subjected to Box–Cox transformation before statistical analysis. The significance threshold was set at 0.05.

## Home cage activity, USVs, and social behavior

All animals had ad libitum access to food and water under a standard 12-h light/dark cycle (7:00 pm—7:00 am) with a regulated

ambient temperature of 22°C ± 1 and at a relative humidity of 40–50%. All experiments were conducted in strict compliance with the National Institutes of Health Guidelines for the Care and Use of Laboratory Animals and approved by the Regional Council in Karlsruhe (Regierungspräsidium Karlsruhe, Germany; Animal Ethic Protocol 35–9185.81/G-104/16 and G-105/16). All experiments were performed by an experimenter blinded to the genotype of mice. For the order of tests, see Appendix.

### Home cage, diurnal and repetitive behavior

Home cage activity was recorded using the LABORAS system (Metris B.V., Netherlands), that allows the automatic detection of behavior-specific vibrations produced by the animal in a home cage that is located on a carbon fiber platform. This allows to distinguish between climbing, grooming, rearing, distance travelled, locomotion, and speed parameters over time or as frequency counts. Each animal was placed individually for 72 h in its home cage with free access to water and food. Recording started after a habituation period of 24 h. Behavioral parameters were calculated using the LABORAS software (version 2.6.2).

### Ultrasonic vocalizations

The animals were habituated to the test room 1 h prior to the recording phase. Seven-day-old pups were separated from the litter and placed in an empty glass container (6 × 9.5 × 7.5 cm) with the floor covered with bedding material. The container was placed in the middle of a box that was covered with a transparent plexiglass lid that held a microphone 30 cm above the pup. The emitted calls were recorded for 5 min with an acquisition setup from Avisoft Bioacoustics (Berlin, Germany) consisting of Ultrasound Gate 416 Hb USB audio device and a CM16/CMPA ultrasonic condenser microphone. The audio signals were digitized at a sampling frequency of 250 kHz with a 16-bit resolution by the Avisoft Bioacoustics RECORDER software (version 4.2.18). For the acoustical analysis, a fast Fourier transformation was performed using Avisoft SASLab Pro software (version 5.2.07; 512 FFT-length, 100% frame Hamming window, and 75%-time window overlap) followed by a spectrogram creation with a 488 Hz frequency resolution and 0.512-ms time resolution. Single calls were detected using the automatic whistle tracking algorithm with a minimum call duration of 3 ms and a hold time mechanism of 10 ms. A cut-off filter of 15 kHz and a postfilter (minimum duration of 1 ms and entropy of 0.5) were applied. The accuracy of tracked calls was verified manually by an experienced user (blind to mouse genotype) to add missed calls or remove false positives due to extraneous noise from the automatically analyzed sonograms, if necessary. The duration and maximum peak frequency of each call, as well as the number of calls per animal, were determined automatically from the obtained sonogram. As qualitative parameters, the call types were classified manually as described (Scattoni *et al*, 2008; Takahashi *et al*, 2016) and their chronology was analyzed to gather information about repetitive behavior.

### Social interaction of adult mice

Social behavior of adult mice (age: 2 months) was analyzed in the 3 chambers test with a protocol adapted from Yang et al (2011). The testing arena consisted of a transparent plexiglass box (20 × 61 × 40 cm) divided into three chambers of equal size that are arranged in a row. Chambers are separated by doors (5 × 8 cm) that can be blocked. For habituation, the test mouse was placed in the middle chamber for 10 min with no access to the side chambers. Next, the doors to both side chambers were opened to allow habituation and exploration of all compartments for 10 min. To test the preference for social novelty versus a novel object, an empty wire cage (novel object) was added to one side chamber and an unfamiliar mouse (stranger 1, same age, same sex, same genetic C57BL6 background) was placed in a second wire cage in the other side chamber. The wire cages (diameter: 7.7 cm; height: 17.5 cm; bars spaced 1 cm apart) allow nose contact between the bars, but prevent fighting. Doors to both side chambers were unblocked, and the test mouse was allowed to explore all compartments freely for 10 min. At the end of the first testing session, each mouse was tested in a second session to quantify social preference. To this end, a second unfamiliar mouse (stranger 2; genetic background: NMRI) was inserted in the previously empty wire cage and the test mouse was observed again for 10 min to analyze preference for social novelty. All mice serving as "stranger mice" were used up to three times per day. All test sessions were recorded with an Eneo VK-1316S camera and analyzed automatically with the SYGNIS Tracker software (version 4.14). The time spend in each chamber was extracted from the video data.

### Olfaction test

The ability for olfaction was assessed in adult mice (age: 2 months) with a buried food test as described (Yang & Crawley, 2009). In this test, mice were placed in individual cages with 15 min to find a piece of chow (Kelloggs® Honey Bsss Loop) that was buried in the bedding. Mice were deprived of food 18–24 h prior to the test with access to water *ad libitum*. Animals were habituated to the test room one hour prior to the recording phase. Then, the test mouse was placed for 5 min in the test cage (without steel top, no food buried) for habituation. Afterward, the mouse was removed from the cage and the food pellet buried 1 cm beneath the surface. Subsequently, the mouse was reintroduced into the test cage and the latency to find and start eating the food pellet was recorded. A latency of 900 s was recorded as a failure to find the pellet.

### Elevated Plus maze

The elevated plus maze allows to analyze anxiety of mice by creating a conflict for the animal between their natural tendency to explore a novel environment or to avoid brightly lit open areas. The arena consists of a plus-shaped elevated platform (gray opaque plastic material; 70 cm above the floor) that has a central intersection (6 × 6 cm) from which the animals can enter four equally sized arms (6 cm width × 35 cm length) freely. Two opposing arms are flanked by opaque walls (17 cm); the remaining arms are open areas without walls. The illumination was adjusted by indirect fluorescent overhead light to 50 lux in the closed arms and 120 lux at the open arms. The animal was set into the central intersection part of the arena and measured for 10 min. All tests were recorded with Version 4.14 camera and analyzed automatically with the SYGNIS Tracker software (version 4.14). The tracked zones include all arms and the intersection area. The parameters extracted from the video are the number of visits and the time spent in either open or closed arms. One LM was excluded from EPM analysis due to seizures.

# Data availability

This study includes no data deposited in external repositories.

**Expanded View** for this article is available online.

## Acknowledgements

We acknowledge funding by the Deutsche Forschungsgemeinschaft (MU 1457/15-1 and MU 1457/17-1 to UM and KO 1674/28-1 to MK). DPW is a member of the Neuroscience Center Zurich (ZNZ) and of the Zurich Center for Integrative Human Physiology (ZIHP). We are grateful to Klaus-Armin Nave (MPIEM Göttingen) for providing NexCre mice. We are grateful to Julia Gobbert and Inger Drescher for excellent technical assistance and for Dr. Claudia Pitzer and Barbara Kurpiers from the Interdisciplinary Neurobehavioral Core for help with behavioral tests. Open Access funding enabled and organized by Projekt DEAL.

## Author contributions

UCM designed and conceived the study. VS performed immunohistochemistry and performed and analyzed experiments of ASD-related behavior including diurnal activity, repetitive behaviors, ultrasonic vocalizations, social interactions, elevated plus maze, and olfaction. SE performed and analyzed neuronal morphology including spine density analysis and interpreted results. DF, MR, and KH performed immunochemistry. SL and MK performed extracellular electrophysiological recordings in brain slices, analyzed data, and interpreted results. MKB and JvE performed patch-clamp recordings, analyzed data, and interpreted results. LS performed stereology, analyzed, and interpreted data on brain anatomy. IA and DPW conducted, analyzed, and interpreted neuromotor and cognitive behavioral experiments. UCM wrote the manuscript with help and input from all authors. All authors approved the final manuscript.

## Conflict of interest

The authors declare that they have no conflict of interest.

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
