## [Review Process File · The EMBO Journal]

Loss of all three APP family members during development impairs synaptic function and plasticity, disrupts learning and causes an autism-like phenotype.

Vicky Steubler, Susanne Klein, Michaela Back, Susann Ludewig, Dominique Fässler, Max Richter, Kang Han, Lutz Slomianka, Irmgard Amrein, Jakob von Engelhardt, David Wolfer, Martin Korte, and Ulrike Muller

DOI: [10.15252/embj.2020107471](https://doi.org/10.15252/embj.2020107471)

Corresponding author(s): [Ulrike Muller \(u.mueller@urz.uni-heidelberg.de\)](mailto:u.mueller@urz.uni-heidelberg.de)

Review Timeline:

Submission Date:	8th Dec 20
Editorial Decision:	29th Jan 21
Revision Received:	9th Mar 21
Editorial Decision:	17th Mar 21
Revision Received:	29th Mar 21
Accepted:	1st Apr 21

Editor: Karin Dumstrei

Transaction Report:

Dear Ulrike,

Thank you for submitting your manuscript to The EMBO Journal. Sorry for the delay in getting back to you with a decision, but I have now received the three comments on your manuscript.

As you can see from the comments below, the referees find the analysis interesting, well done and support publication here. They raise a number of different concerns that should be fairly straight forward to address. Let me know if we need to discuss any further specifics.

When preparing your letter of response to the referees' comments, please bear in mind that this will form part of the Review Process File, and will therefore be available online to the community. For more details on our Transparent Editorial Process, please visit our website:

<https://www.embopress.org/page/journal/14602075/authorguide#transparentprocess>

Thank you for the opportunity to consider your work for publication. I look forward to your revision.

with best wishes

Karin

Karin Dumstrei, PhD
Senior Editor
The EMBO Journal

The revision must be submitted online within 90 days; please click on the link below to submit the revision online before 29th Apr 2021.

Referee #1:

Review of EMBOJ-2020-107471

Steubler et al investigated the effects of knocking out the amyloid precursor protein (APP) family during development in excitatory forebrain neurons on neuroanatomy, synaptic transmission, learning and memory, and autism spectrum disorder (ASD) like phenotypes. The authors utilize a vast array of histological, electrophysiological, and behavioral tests to thoroughly explore the developmental effects of the APP family deletion. Authors report corpus callosum and hippocampal architecture defects, independent of cortical structural effects. Authors also demonstrate a complex and layer dependent impairment in CA1 dendritic structure. Concordantly, an electrophysiological examination revealed altered pre-synaptic release dynamics and resulting layer specific mini excitatory and inhibitory post synaptic potentials (mE/I PSPs). Authors also demonstrated altered input/output and significantly reduced field EPSPs (fEPSPs) and a subsequent failure to induce or maintain long term potentiation. The altered EPSPs were correlated with a strong impairment in learning and memory tasks. Authors also noted a stark phenotype of hyperactive, repetitive, and perseverative behaviors. Analysis of ultrasonic vocalizations (USVs), home cage hyperactivity, and social behaviors also indicated an ASD phenotype in the APP cTKO mice.

This manuscript does an excellent job of describing the in-depth phenotypic result of deleting the APP family in late development. The decreased apical dendrite complexity and altered spine dynamics highlight a key role for APP family members in hippocampal assembly. This is an important finding in the heretofore undefined role of the APP family in cognitive development. The novel report of deep versus superficial CA1 cells is also quite exciting and may be relevant to disease phenotypes for APP family diseases. The ASD like phenotype (stereotypic non-repetitive behavior) found in the developmental cTKO and the altered electrophysiology are both important steps in understanding the role of the APP family in development, DS, and AD. The work presented in this manuscript is both novel and a step in determining the functional effects of an APP cTKO, however, a few concerns and questions arise from the manuscript:

- Given the susceptibility of animals to seizure, was seizure activity in these animals explored?
 - o Was the animal noted to have a seizure excluded from the electrophysiology following exclusion from behavior? If not, the animal should be removed.
- Is there an altered survival curve of the cTKO?
- The calbindin expression should be shown in Figure 1 as the distinction between superficial and deep pyramidal cells is key to the morphology differences
- The total pyramidal cell counts should be graphed and displayed in Figure 1
- Figure 1B requires representative images.
- The WT group should be included in all of Figure 1, to validate the control in these experiments.
- Spine density is not reflective of only excitatory synapses. Only certain subtypes of spines indicate excitatory mature synaptic connections, whereas others are reflective of immature and transient spines. An analysis of mature vs immature spines, or histology would be appreciated to validate the claim of excitatory synapse changes.
- An examination of the mechanism of LTP failure would be appreciated? For example, was LTP failure replicated with other potentiation techniques (HFS/chemical) and was LTD also uninducible? Additionally, were there quantifiable changes in NMDA/AMPA/mGLUR receptor levels in these animals?
- The search strategy of the cTKO mice would be better understood with a representative path or a heat map. This would help understand the quadrant preference and if most of the time was spent at walls or in the starting quadrant.
- Supplemental Figure 4C shows difference in anxiolytic behavior with arm entries and 4B shows a possible difference in time spent in the light quadrant. It may be helpful to elaborate on this in the text, or perform a different anxiety test, such as fear conditioning or light-dark box.

Minor concerns:

There is a typo in the legend for figure 6. And 6D should be relabeled for clarity.

Referee #2:

Since the first molecular cloning of APP more than 30 years ago and the identification of APP mutations linked to familial forms of AD there has been much work on the role of APP and A β for AD etiology but analysis of its physiological function remain sparse. In the present paper the authors characterized a newly generated conditional knock out mouse line NexCrecTKO that lack the APP family in excitatory forebrain neurons from embryonic day 12.5 days on. They find an essential role of the APP family for synaptogenesis, synaptic plasticity and learning and social behaviors. Specifically, NexCrecTKO mice had reduced synaptic transmission, reduced long-term potentiation that was associated with reduced spine density and dendritic length. Furthermore, those mice showed several impairments in a variety of different behavior tests including learning and memory, repetitive rearing and climbing and social interaction suggesting essential functions for APP in normal hippocampal function. The authors have used a variety of methods to substantiate this claim and all the results speak in the same direction. This is an important paper which illuminates the physiological role of APP and will clearly be of great interest to a wide readership. The manuscript is well-written and the conclusions right to the point. I recommend it to be published in EMBO J though I have some minor changes:

- On page 9 last sentence before the new headline: "We conclude that the overall impaired basal synaptic transmission.....". At the end of this sentence, a dot is missing. Please punctuate the end

of the sentence.

- In the result section the order of the figures are sometimes not in a sequential arrangement e.g. on page 10 first paragraph the results of Fig. 6 are presented first and then on page 11 Fig. 5 follows. Instead, the results should be presented in a logical sequence.
- On page 13 there is a fragmentary sentence that I do not understand: "As body weight of pups was comparable between genotypes (LM:.....)." Please complete the sentence.
- The discussion seems a little bit long and can be shortened.

Referee #3:

This work extends earlier, extensive work of the senior author and her colleagues in understanding the role of the APP family in normal brain function and in disease. Using an APLP1 complete knockout as background strain, APP and APLP2 were conditionally knocked-out in excitatory neurons from E12.5 onwards. Adult mice were analysed histologically and at the level of the synapse, by performing a Scholl analysis and a range of electrophysiological recordings as well as a very extensive behavioural test battery that also allowed for the analysis of autism-like behaviours (breeding, nesting, repetitive behaviours, social communication and ultrasonic vocalisation). The study demonstrates an essential role for the APP family specifically in the hippocampus.

General comments:

The study has generally been well done and the manuscript is easy to follow.

(1) What in my view would help is a scheme added to Figure 1 of the three main players and their domains, as well as a simple scheme of how the TKO strain has been generated.

(2) What would also help is a study design figure panel that provides information about ages (page 6 talks about young and aged mice but it is not clear what that is) and the order in which the different types of analyses were done (e.g. the order of the behavioural tests) and the n number of mice involved in any given study. Has the behavioural analysis been done with all mice and the ephys with a subset and the histology with another subset?

(3) Considering the poor breeding behaviour of the TKO strain a breeding scheme (in the supplement) would be helpful. Is there embryonic lethality which remains unobserved? Is there a genetic difference between the strains considering that some mouse strains partially lack the corpus callosum?

(4) Comments regarding the figures:

The figures are not referenced in the text chronologically. 1D appears before 1A; 2B,C are not referenced at all, and the contents of 3F would ideally been shown in Fig 2.

Fig 1A looks cut-off, the red arrowheads are not referenced in the legend and one black arrowhead is at the edge of the image. In my view, it would be better to replace the TKO image in 1A and show two images, one with mild dysgenesis and one with total agenesis. Fig 1D lacks a unit (caudal-> rostral), and the DAPI label is difficult to read in 1C.

Fig 2A is flipped in its orientation in relation to Fig 1A. The individual data points (black circles) are difficult to see against the blue bar (2E,F,H, 3A,B, 4 etc).

Additional minor comments:

- (5) last sentence page 5: '...ectopias in the marginal zone of the cortex.' Maybe add an explanation, starting with '..., indicating that...'
- (6) Possibly edit sentence: 'layer identity along the radial axis ...was not affected...'. Does this mean not disrupted?
- (7) I would suggest: Last paragraph page 6: change 'to the number of sections' to 'to the number of x um sections'. Change (> 9 sections) to (> 9 of x? sections)
- (8) Page 6, Fig 1C: to me the calbindin pattern looks also different. Further down the page it says 'data not shown' - I suggest showing the data.
- (9) Page 6, 2nd paragraph. Full stop missing.
- (10) Fig 2A, dCA1 dark blue - this looks to me rather like magenta.
- (11) Discussion: regarding the role of the APP family. Is there any ortholog information from invertebrates such as *C. elegans* that would illuminate the role of APP?

Replies to reviewer comments on Steubler et al., EMBOJ-2020-107471

We would like to thank all three reviewers for their overall very positive and constructive comments, which helped us to greatly improve the manuscript. All three reviewers found the paper very interesting, judged the work as important and novel and stated that we did an excellent job in providing an in-depth phenotypic analysis.

Our point-to-point replies are as follows. We would also like to point out, that we include new data in: Figure 1A,C,D and in EV1; in Figure 4K,L; and in Figure 6R. In addition, we now provide a table describing the study design (age and number of animals used for the specific experiments) as an appendix. We also modified Figure 2, 3 and 4 as suggested. To facilitate tracking of changes in our revised manuscript we have highlighted these changes in the manuscript (in yellow).

Referee #1:

General remark: we thank the reviewer for his/her overall very positive judgement of our work indicating that “the manuscript does an excellent job of describing the in-depth phenotypic result of deleting the APP family in late development.”

Point 1: Given the susceptibility of animals to seizure, was seizure activity in these animals explored?

Reply: So far, we have noted seizures in one cTKO animal when performing behavioral experiments. This animal was therefore excluded from the analysis, as indicated in the methods section. We agree with the reviewer that it may be interesting to explore seizure activity in more detail in future studies.

Point 2: Was the animal noted to have a seizure excluded from the electrophysiology following exclusion from behavior? If not, the animal should be removed.

Reply: The electrophysiology was not performed with animals that underwent prior behavioral testing but with independent sets of animals (see also appendix for an overview).

Point 3: Is there an altered survival curve of the cTKO?

Reply: We checked the survival of juvenile animals at weaning (see new Figure EV1B). For this we genotyped offspring obtained after in vitro fertilization (IVF) and implantation into wild type foster mothers. At weaning (3 weeks of age) genotype distribution did not differ significantly from the expected mendelian frequency of 50% ($\chi^2(1,N=337)=1.02$, $p=0.3132$, ns), excluding embryonic or early postnatal lethality.

Adult animals: as cTKO mice were impaired in mating, most animals used in this study were generated by IVF to be used in specific experiments at the age of 4-6 months. A separate small group of 5 LM and 5 cTKO mice was used for brain anatomy at the age 18-20 months (see appendix). For these time points we did not observe any obvious difference in viability between genotypes. As only small cohorts of mice were used for the various experiments, we cannot formally exclude, however, some nonobvious degree of lethality after weaning.

Point 4: The calbindin expression should be shown in Figure 1 as the distinction between superficial and deep pyramidal cells is key to the morphology differences.

Reply: as suggested, we now show IHC for Calbindin as a new panel in Figure 1C. The manuscript now reads: “The expression of calbindin, as one marker expressed by superficial CA1 cells (Soltesz & Losonczy 2018) was, despite the delamination, not affected in NexCre cTKOs. Superficial CA1 cells of cTKO mice still expressed calbindin while deep CA1 cells did not (Fig 1C, right panel)”.

Point 5: The total pyramidal cell counts should be graphed and displayed in Figure 1

Reply: as suggested CA1 pyramidal cell counts of WT, LM and cTKO are now shown as a new panel in Figure 1D.

Point 6: Figure 1B requires representative images.

Reply: The quantification of callosal agenesis (Fig 1E) involves serial coronal sections that are inspected for the presence or absence of the corpus callosum (CC) along the rostral-caudal axis. To make this clearer for the reader we now provide a scheme derived from the Allen brain atlas depicting the CC in coronal sections (see EV1C). In WT mice the CC extends approximately from bregma +2.0 to bregma -2.5 and was observed in 11-13 consecutive sections in this study. For bregma 1,9 we provide in Fig 1C representative images for a WT animal (normal CC), a LM (normal CC) and a cTKO mouse (agenesis of the CC). In addition, we have improved the labelling of Fig 1C to indicate more clearly the CC.

The revised manuscript now reads: “In WT and LM control mice, the CC was observed in 11 to 13 consecutive coronal sections. The corpus callosum was assessed and rated, according to the number of sections (see scheme in Fig EV1C) in which the CC was observed as normal (>9 consecutive sections), mildly dysgenic (7 - 9 sections), severely dysgenic (< 7 sections) or agenic (0 sections). None of the littermate or WT controls showed a dysgenic CC (Fig 1C,E). Agenesis of the CC was found in the majority of NexCre cTKOs, the remainder being severely dysgenic with only one, mildly dysgenic exception (Fig 1E).

Point 7: The WT group should be included in all of Figure 1, to validate the control in these experiments.

Reply: We now include WT animals in Figure 1A,C and D and quote a paper (Soltesz & Losonczy 2018) that Calbindin is a typical marker of superficial CA1 cells in WT mice.

Point 8: Spine density is not reflective of only excitatory synapses. Only certain subtypes of spines indicate excitatory mature synaptic connections, whereas others are reflective of immature and transient spines. An analysis of mature vs immature spines, or histology would be appreciated to validate the claim of excitatory synapse changes.

Reply: We agree with the reviewer that it is interesting to assess the distribution of mature versus immature/transient spines in cTKO mice in comparison to controls. To this end, the reviewer suggested to perform histology. However, this is technically not possible using immunohistochemistry (IHC), as dendrites of superficial sCA1 and deep dCA1 cells cannot be distinguished by IHC. For example, only apical dendrites of sCA1 cells show reduced spine

density, whereas apical dendrites of dCA1 cells (that would be detected as well) show normal spine density. Moreover, IHC will not be sensitive enough to reliably detect differences in spine density in the range of 12-15%, as observed in cTKO mice.

Although we consider the reviewer's suggestions very interesting, we count on the kind understanding of this reviewer that tackling these aspects in detail warrants further investigation (e.g. looking at spine types and their dynamics, evaluate spine head size etc.) that we feel is beyond the scope of the present manuscript and that we would like to address in a future study. Nevertheless, to make this point clear to the reader we modified the discussion accordingly and write: "It will also be interesting to investigate whether cTKO mice show possible differences in the relative abundance of mature versus immature spines and their turnover, as APP-KO mice showed impairments in environmental enrichment induced spine plasticity within the cortex (Zou et al, 2016)".

Point 9: An examination of the mechanism of LTP failure would be appreciated? For example, was LTP failure replicated with other potentiation techniques (HFS/chemical) and was LTD also uninducible? Additionally, were there quantifiable changes in NMDA/AMPA/mGLUR receptor levels in these animals?

Reply: We would like to point out that we found LTP being severely reduced in cTKOs. Nevertheless, LTP can still be induced at low level (significantly above baseline) by theta burst stimulation in cTKO slices. Therefore, we do not consider the reported LTP responses as a complete LTP failure. Theta burst stimulation depends on the activation of NMDARs (not mGLURs). We believe that other stimulation (HFS) techniques are unlikely to yield major additional insights. Regarding LTD, which is also dependent on NMDARs, we consider this as extremely difficult to address, as we have repeatedly failed to record LTD responses in hippocampal slices from adult animals.

As suggested by the reviewer and in order to address the mechanism of impaired LTP in more detail, we now performed additional experiments and assessed NMDAR expression and functionality in cTKO slices (see new Fig 4K,L). To this end we recorded AMPA- and NMDA receptor-mediated currents at a holding potential of -70 and +40 mV, respectively (Fig 4K,L). The ratio of AMPA/NMDA receptor-mediated currents (A/N) in cTKO neurons did, however, not significantly differ from that obtained in wild type or LM neurons, suggesting that the synaptic content of AMPA receptors and NMDA receptors is not grossly altered in cTKOs (Fig 4K,L).

Mechanism of impaired LTP

As patch clamp analysis of CA1 cells yielded only rather minor alterations of synaptic changes at the level of individual CA1 cells, including no gross alteration in the functionality of NMDA receptors, we consider impaired associativity as the most likely explanation for impaired LTP. We have modified the discussion accordingly and added the following paragraph on p14: "LTP is a process whereby brief periods of synaptic activity, like TBS used here, can produce a long-lasting increase in synapse strength. There are three properties of LTP: (1) cooperativity,

(2) input specificity, and (3) associativity, which are essential for learning and memory in mammals (Kandel et al, 2013). Associativity means that co-activation of weak inputs with strong inputs onto the same neuron can strengthen the weak input. In our case, NMDAR function is not grossly altered, but LTP is compromised and the number of inputs to a CA1 pyramidal neurons is reduced, which most likely reduces the power of associativity and cooperativity. Without altering the functional properties of a single synapse this would already lower the probability to induce LTP and is the most likely explanation for the LTP deficits observed in cTKO mice.“

Point 10: The search strategy of the cTKO mice would be better understood with a representative path or a heat map. This would help understand the quadrant preference and if most of the time was spend at walls or in the starting quadrant.

Reply: Thanks for this suggestion. We now provide representative occupancy plots during the probe trial (new Figure 6R). In the Legend we write: “Dwell times were summed across all tested LM and cTKO mice, respectively, and then z-transformed. LM mice show a highly focused spatial search strategy for the trained platform position, while cTKO spend most of their time close to the wall”.

Point 11: Supplemental Figure 4C shows difference in anxiolytic behavior with arm entries and 4B shows a possible difference in time spent in the light quadrant. It may be helpful to elaborate on this in the text, or perform a different anxiety test, such as fear conditioning or light-dark box.

Reply: We thank the reviewer to point this out. We have modified the revised manuscript to explain the behavior of cTKO mice in the elevated plus maze in more detail. The revised manuscript now reads on p14. “While LM mice showed a very pronounced preference for the closed arms of the EPM, the behavior of cTKOs was considerably more variable (Fig EV5, B), pointing towards a possible anxiolytic phenotype. Despite this, cTKO still spend, on average, significantly more time in the closed arms, to an extent that was statistically comparable to LM controls. This more variable behavior was also reflected when analyzing the number of visits into open versus closed arms (Fig EV5C). cTKO mice visited open and closed arms with similar frequency, although the mean increase in the number of visits to open arms was largely due to 4 animals. This apparent increase in risk taking behavior observed for some of the cTKOs contrasts, however, with their aversion to novelty, as seen by increased freezing in the start zone of the Barnes maze (Fig 6K)“.

Minor concerns:

There is a typo in the legend for figure 6. And 6D should be relabeled for clarity.

Reply: Thanks for pointing this out. We relabeled previous Fig 6D (now 5D). The heading now reads: “Communication: Ultrasonic vocalization in pubs”. The Y-axis of Fig 5D was relabeled to “Number of calls in 3 min”.

Referee #2

General remark: we thank the reviewer for his/her overall very positive judgement of our work stating “This is an important paper which illuminates the physiological role of APP and will clearly be of great interest to a wide readership. The manuscript is well-written and the conclusions right to the point”.

Point 1: On page 9 last sentence before the new headline: "We conclude that the overall impaired basal synaptic transmission.....". At the end of this sentence, a dot is missing. Please punctuate the end of the sentence.

Reply: Thanks for pointing this out, the sentence now ends with a full stop.

Point 2: In the result section the order of the figures are sometimes not in a sequential arrangement e.g. on page 10 first paragraph the results of Fig. 6 are presented first and then on page 11 Fig. 5 follows. Instead, the results should be presented in a logical sequence.

Reply: Thanks for pointing this out. We relabeled the figure panels strictly as they are mentioned in the text. Accordingly, we start on page 10 with the description of homecage activity, former Fig 6A, which is now referenced as Fig 5A.

Point 3: On page 13 there is a fragmentary sentence that I do not understand: "As body weight of pups was comparable between genotypes (LM:.....)." Please complete the sentence.

Reply: The sentence now reads: “As body weight of pups was comparable between genotypes (LM: 5.65 ± 0.11 g vs NexCre cTKO 5.33 ± 0.15 g, unpaired Student’s t-test: ns) this excludes major differences in overall strength as a cause of impairments”.

Point 4: The discussion seems a little bit long and can be shortened.

Reply: we have deleted a few sentences to shorten the discussion, but trust on the kind understanding of the reviewer that the overall length of the manuscript is similar, due to adding a paragraph on the mechanism of impaired LTP (page 19), as requested by reviewer#1.

Referee #3

General remark: we thank the reviewer for his/her overall very positive judgement of our work stating that “The study has generally been well done and the manuscript is easy to follow.”

Point 1: What in my view would help is a scheme added to Figure 1 of the three main players and their domains, as well as a simple scheme of how the TKO strain has been generated.

Reply: As suggested by the reviewer we now provide a scheme (see new Figure EV1A) depicting the domains of APP, APLP1 and APLP2 and in addition as new Figure EV1B a scheme depicting the breeding strategy

Point 2: What would also help is a study design figure panel that provides information about

ages (page 6 talks about young and aged mice but it is not clear what that is) and the order in which the different types of analyses were done (e.g. the order of the behavioural tests) and the n number of mice involved in any given study. Has the behavioral analysis been done with all mice and the ephys with a subset and the histology with another subset?

Reply: As suggested we now provide a table (see appendix) indicating the age and number of animals that were used for the specific experiments. From the color code it is now also clear which experiments were performed with the same set of animals. The order of cognitive tests had already been indicated in the methods section and is now also indicated in the appendix. Specifically, the electrophysiology was done with two separate groups of animals that did not undergo previous behavioral testing. The histology was performed using another separate set of animals.

Regarding the aged animals mentioned on page 6: we now specify the age (18-20 months) in the text. Counting of CA1 cells was performed using the same animals as used for stereology. We re-checked the revised manuscript to make sure that ages and numbers of animals are always indicated in the figure legends and/or in the text.

Point 3: Considering the poor breeding behaviour of the TKO strain a breeding scheme (in the supplement) would be helpful. Is there embryonic lethality which remains unobserved?

Reply: We now provide a breeding scheme in Fig EV1B, as suggested. We checked the survival of juvenile animals at weaning (see new Figure EV1B). For this we genotyped offspring obtained after in vitro fertilization (IVF) and implantation into wild type foster mothers. At weaning (3 weeks of age) genotype distribution did not differ significantly from the expected mendelian frequency of 50% ($\chi^2(1, N=337)=1.02$, $p=0.3132$, ns), excluding embryonic or early postnatal lethality.

Is there a genetic difference between the strains considering that some mouse strains partially lack the corpus callosum?

Reply: The genetic background is highly similar between WT, LM and cTKO mice. The mutant mouse lines with modifications of only one family member (APPflox, APLP2flox, APLP1-KO) had previously been backcrossed to C56BL6 for more than 6 generations. Importantly, C56BL6 mice (see new Figure 1C) and LM controls of the same genetic background show no callosal agenesis, whereas cTKO do. In contrast, WT mice of 129 strains, as correctly pointed out by the reviewer, have previously been reported to show up to 30% callosal agenesis

Lipp, H.-P. & Wahlsten, D. (1992) in *Absence of the Corpus Callosum*, ed. Driscoll, P. (Birkhäuser, Basel), pp. 217–252.). Thus, our analysis did not involve a genetic background that predisposes to callosal agenesis

Point 4a: Comments regarding Figure 1:

The figures are not referenced in the text chronologically. 1D appears before 1A;

Fig 1A looks cut-off, the red arrowheads are not referenced in the legend and one black arrowhead is at the edge of the image. Fig 1D lacks a unit (caudal-> rostral), and the DAPI label is difficult to read in 1C.

Reply: We thank the reviewer for pointing this out and apologize that this has been overlooked. We now re-arranged the figure panels strictly as they are mentioned in the text. We also

improved the labelling of the histology with white arrowheads pointing to ectopic cells and red arrowheads that indicate the abnormal CA3 cell layer. Further, we now indicate more clearly the corpus callosum. As requested we now explain the unit-less labelling of the x-axis (Figure 1A) describing the volumetric measurements along the rostral-caudal axis.

The legend of Fig 1A reads: “Stereological evaluation of cortical volume of young (Y-WTC, Y-LMC, Y-cTKO, age: 5-6 months, n=5) and old (O-LM n=5, O-cTKO n=6, age: 18-20 months) mice in 12 equidistant bins along the caudal-rostral axis of the coronally section brain. Each bin would comprise an approximately 570 μm thick slab of the brain”.

Point 4b: In my view, it would be better to replace the TKO image in 1A and show two images, one with mild dysgenesis and one with total agenesis.

Reply: The quantification of callosal agenesis (Fig 1E) involves serial coronal sections that are inspected for the presence or absence of the CC along the rostral-caudal axis. To make this clearer for the reader we now provide a scheme depicting the CC in coronal sections (see EV1C). In WT mice the CC extends approximately from bregma +2.0 to bregma -2.5 and was observed in 11-13 consecutive sections in this study. For bregma 1.9 we provide in Fig 1C representative images for a WT animal (normal CC), a LM (normal CC) and a cTKO mouse (agenesis of the CC). In addition, we have improved the labelling of Fig 1C to indicate more clearly the CC.

The revised manuscript now reads: “In WT and LM control mice, the CC was observed in 11 to 13 consecutive coronal sections. The corpus callosum was assessed and rated, according to the number of sections (see scheme in Fig EV1C) in which the CC was observed as normal (>9 consecutive sections), mildly dysgenic (7 - 9 sections), severely dysgenic (< 7 sections) or agenic (0 sections). None of the littermate or WT controls showed a dysgenic CC (Fig 1C,E). Agenesis of the CC was found in the majority of NexCre cTKOs, the remainder being severely dysgenic with only one, mildly dysgenic exception (Fig 1E).

Point 4c: Comments regarding Figure 2: 2B,C are not referenced at all, and the contents of 3F would ideally have been shown in Fig 2. Fig 2A is flipped in its orientation in relation to Fig 1A.

Reply: We thank the reviewer to point this out. All panels are now referenced in the text. We also modified Figure 2B and flipped its orientation, in order to display the same orientation in all figure panels. The scheme depicting the Sholl spheres is now shown in Fig 2B, as suggested.

Point 4d: The individual data points (black circles) are difficult to see against the blue bar (2E,F,H, 3A,B, 4 etc).

Reply: We have improved the contrast and now display the individual data points as white circles in the bar graphs of all panels (Figure 2, 3 and 4).

Additional minor comments:

Minor Point 5: last sentence page 5: ...ectopias in the marginal zone of the cortex.' Maybe add an explanation, starting with '...', indicating that...'

Reply: As requested we have improved this sentence that now reads: “In previously generated constitutive triple KO mice (Herms et al, 2004), we had observed focal neuronal ectopias in the marginal zone of the cortex, indicating overmigration of neuroblasts.”

Minor Point 6: Possibly edit sentence: 'layer identity along the radial axis ...was not affected...' Does this mean not disrupted?

Reply: As this may not have been sufficiently clear we now rephrased the statement to: “The expression of calbindin, as one marker expressed by superficial CA1 cells (Soltesz & Losonczy 2018) was, despite the delamination, not affected in NexCre cTKOs. Superficial CA1 cells of cTKO mice still expressed calbindin while deep CA1 cells did not (Fig 1C, right panel)”.

Minor Point 7: I would suggest: Last paragraph page 6: change 'to the number of sections' to 'to the number of x um sections'. Change (> 9 sections) to (> 9 of x? sections)

Reply: please see also our reply to point 4. We have also addressed this issue by improving the legend of Fig 1A which now reads: “Stereological evaluation of cortical volume of young (Y-WTC, Y-LMC, Y-cTKO, age: 5-6 months, n=5) and old (O-LM n=5, O-cTKO n=6, age: 18-20 months) mice in 12 equidistant bins along the caudal-rostral axis of the coronally section brain. Each bin would comprise an approximately 570 µm thick slab of the brain”.

Minor Point 8: Page 6, Fig 1C: to me the calbindin pattern looks also different. Further down the page it says 'data not shown' - I suggest showing the data.

Reply: We now show the data for Calbindin staining within the CA1 layer in Figure 1C, right panel (see also our reply to Minor point 6).

We agree with the reviewer that the overall intensity of the Calbindin staining depicted in Figure 1B is slightly reduced in LM sections, as compared to cTKO sections. However, this difference is within the range of variability also observed in groups of identical genotype. We wanted to illustrate that the overall pattern and width of layers is comparable between LMs and cTKOs, such that Calbindin is confined to layers II/III and V which indicates that cortical layer formation is not obviously disturbed.

Minor Point 9: Page 6, 2nd paragraph. Full stop missing.

Reply: We thank the reviewer to point this out.

Minor Point 10: Fig 2A, dCA1 dark blue - this looks to me rather like magenta.

Reply: We hope that the reviewer agrees with our decision to keep the color designation. The impression of magenta may be related to differences in color displayed by various computer screens or printers.

Minor Point 11: Discussion: regarding the role of the APP family. Is there any ortholog information from invertebrates such as *C. elegans* that would illuminate the role of APP?

Reply: The discussion is already rather long and we believe that a more detailed discussion of ortholog functions is beyond the scope of this study, that focuses on the mammalian APP family. Instead, we now mention orthologues in the introduction and refer the reader to several references for more detailed information on that topic.

The manuscript now reads: “APP is a type I single pass transmembrane protein that belongs to an evolutionary conserved gene family including APL-1 in *C. elegans*, APPL in *Drosophila*, Appa and Appb in *zebrafish* and in mammals besides APP the amyloid precursor like proteins APLP1 and APLP2 (reviewed in (Müller et al, 2017); for recent examples on studies in orthologs see (Banote et al, 2020; Ewald & Li, 2012; Kessissoglou et al, 2020; Rieche et al, 2018; Wang et al, 2017)).“

Dear Ulrike,

Thank you for submitting your revised manuscript to The EMBO Journal. Your study has now been re-reviewed by referee #1. As you can see from the comments below, the referee appreciates the introduced changes.

I am therefore very pleased to let you know that we will accept the manuscript for publication here. Before I can send you the formal acceptance letter we just need to sort out a few issues.

You can only have 5 keywords - you have at the moment 7

I see that you are missing a Data Availability section. This is the place to enter accession numbers etc. As far as I can see no data is generated that needs to be deposited in a database. If this is correct please state: This study includes no data deposited in external repositories. Please place it after the Materials and methods and before Acknowledgements

Please double check the reference format. There are some citations with more than 10 authors listed.

We don't allow data not shown (pg 6). Please re-phrase or add the data

Please check figure callouts in the text to Fig 3B, C+F panels and Fig 5 B-J panels

The appendix should have a ToC. Please also see guide to authors for how to call out the table in the appendix <https://www.embopress.org/page/journal/14602075/authorguide>

We include a synopsis of the paper (see <http://emboj.embopress.org/>). Please provide me with a general summary statement and 3-5 bullet points that capture the key findings of the paper.

We also need a summary figure for the synopsis. The size should be 550 wide by [200-400] high (pixels). You can also use something from the figures if that is easier.

I have asked our publisher to do their checks on the paper. They will send me the file within the next few days. Please wait to upload the revised version until you have received their comments.

That should be all! You can use the link below to upload the revised version.

Congratulations on a super nice study

with best wishes

Karin

Karin Dumstrei, PhD
Senior Editor
The EMBO Journal

- a point-by-point response to the referees' comments, with a detailed description of the changes made (as a word file).

- a word file of the manuscript text.

- individual production quality figure files (one file per figure)

- a complete author checklist, which you can download from our author guidelines (<https://www.embopress.org/page/journal/14602075/authorguide>).

- Expanded View files (replacing Supplementary Information)

Further information is available in our Guide For Authors:

The revision must be submitted online within 90 days; please click on the link below to submit the revision online before 15th Jun 2021.

Referee #1:

The authors have adequately revised the manuscript.

Manuscript EMBOJ-2020-107471R

Dear Karin,

We were very pleased that referee #1 appreciated the introduced changes and that the manuscript is ready to be accepted.

Below I will briefly summarize the response to your email dated March 17th.

- We reduced the number of keywords to five.
- Our manuscript contains no data that need to be deposited in a database. As you suggested, we included a Data Availability section after the Materials and Methods section that states: „This study includes no data deposited in external repositories“.
- We re-checked the references and do not list more than 10 authors.
- We re-phrased page 6 to avoid the term “data not shown”.
- We checked the figure callouts in the text, all panels of figure 3 are now referenced. Please note that panels B-J of Figure 5 are referenced on pages 13-14.
- We updated the appendix according to your suggestion and added a ToC.
- We generated a summary Figure for the synopsis, a general summary statement and bullet points summarizing the major findings.
- We also checked and responded to the questions to the Figures and Figure legends raised by the publisher. If necessary, we changed the figures according to the suggestions.

Hoping that our manuscript will now be suitable for publication in *The EMBO Journal* and looking forward to your reply,

with best wishes,

Ulrike

Dear Ulrike,

Thank you for submitting your revised manuscript to the EMBO Journal. I have now had a chance to take a look at the introduced changes and all looks good!

I am therefore very pleased to accept the manuscript for publication here.

Congratulations on a nice study.

With best wishes

Karin

Karin Dumstrei, PhD
Senior Editor
The EMBO Journal

Please note that it is EMBO Journal policy for the transcript of the editorial process (containing referee reports and your response letter) to be published as an online supplement to each paper. If you do NOT want this, you will need to inform the Editorial Office via email immediately. More information is available here: https://emboj.embopress.org/about#Transparent_Process

Your manuscript will be processed for publication in the journal by EMBO Press. Manuscripts in the PDF and electronic editions of The EMBO Journal will be copy edited, and you will be provided with page proofs prior to publication. Please note that supplementary information is not included in the proofs.

Should you be planning a Press Release on your article, please get in contact with embojournal@wiley.com as early as possible, in order to coordinate publication and release dates.

If you have any questions, please do not hesitate to call or email the Editorial Office. Thank you for your contribution to The EMBO Journal.

Corresponding Author Name: Ulrike Müller

Journal Submitted to: The EMBO Journal

Manuscript Number: EMBOJ-2020-107471